# Learning Theory Can (Sometimes) Explain Generalisation in Graph Neural Networks

**Pascal Mattia Esser**
Technical University of Munich
esser@in.tum.de

**Leena C. Vankadara**
University of Tübingen
leena.chennuru-vankadara@uni-tuebingen.de

**Debarghya Ghoshdastidar**
Technical University of Munich
Munich Data Science Institute
ghoshdas@in.tum.de

## Abstract

In recent years, several results in the supervised learning setting suggested that classical statistical learning-theoretic measures, such as VC dimension, do not adequately explain the performance of deep learning models which prompted a slew of work in the infinite-width and iteration regimes. However, there is little theoretical explanation for the success of neural networks beyond the supervised setting. In this paper we argue that, under some distributional assumptions, classical learning-theoretic measures can sufficiently explain generalization for graph neural networks in the transductive setting. In particular, we provide a rigorous analysis of the performance of neural networks in the context of transductive inference, specifically by analysing the generalisation properties of graph convolutional networks for the problem of node classification. While VC Dimension does result in trivial generalisation error bounds in this setting as well, we show that transductive Rademacher complexity can explain the generalisation properties of graph convolutional networks for stochastic block models. We further use the generalisation error bounds based on transductive Rademacher complexity to demonstrate the role of graph convolutions and network architectures in achieving smaller generalisation error and provide insights into when the graph structure can help in learning. The findings of this paper could re-new the interest in studying generalisation in neural networks in terms of learning-theoretic measures, albeit in specific problems.

## 1  Introduction

Neural networks have found tremendous success in a wide range of practical applications and, in the broader society, it is often considered synonymous to machine learning. The rapid gain in popularity has, however, come at the cost of interpretability and reliability of complex neural network architectures. Hence, there has been an increasing interest in understanding generalization and other theoretical properties of neural networks in the theoretical machine learning community (Feldman 2020; Arora et al. 2019a; Ma et al. 2017; Nagarajan et al. 2019; Theisen et al. 2020; Ghorbani et al. 2020). Most of the existing theory literature focuses on the supervised learning problem, or more precisely, the setting of inductive inference. In contrast, there is a general lack of understanding of transductive problems, in particular the role of unlabeled data in training. Consequently there has also been little progress in rigorously understanding one of widely used tools for transductive inference—Graph neural networks (GNN).

35th Conference on Neural Information Processing Systems (NeurIPS 2021).

**Graph neural networks.** GNNs were introduced by Gori et al. (2005) and Scarselli et al. (2009), who used recurrent neural network architectures, for the purpose of transductive inference on graphs, that is, the task of labelling all the nodes of a graph given the graph structure, all node features and labels for few nodes. Broadly, GNNs use a combination of local aggregation of node features and non-linear transformations to predict on unlabelled nodes. In practice, the exact form of aggregation and combination steps varies across architectures to solve domain specific tasks (Kipf et al. 2017; Bruna et al. 2014; Defferrard et al. 2016; Veličković et al. 2018; Xu et al. 2019). While some GNNs focus on the transductive setting, sometimes referred to as semi-supervised node classification,[1] GNNs have also found success in supervised learning, where the task is to label entire graphs, in contrast to labelling nodes in a graph. While the understanding of GNNs is limited, there are empirical approaches to study GNNs in the transductive (Bojchevski et al. 2018) and supervised setting (Zhang et al. 2018; Ying et al. 2018). For an extensive survey on the state of the art of GNNs see for example Wu et al. (2020).

**Leaning theoretical analysis of GNNs.** While empirical studies provide some insights into the behaviour of machine learning models, rigorous theoretical analysis is the key to deep insights into a model. The focus of this paper is to provide a learning-theoretic analysis of generalisation of GNNs in the transductive setting. Vapnik first studied the problem of transductive inference and provided generalisation bounds for empirical risk minimization (Vapnik 1982; Vapnik 1998). Subsequent works further analysed this setting in transductive regression (Cortes et al. 2007), and derive VC Dimension and Rademacher complexity for transductive classification (Tolstikhin et al. 2016; El-Yaniv et al. 2009). Generalisation error bounds for 1-layer GNNs have been derived in transductive setting based on algorithmic stability (Verma et al. 2019). In contrast, the focus of the current paper is on learning-theoretic measures, which have been previously used to analyse GNNs in a supervised setting. In Scarselli et al. (2018), VC Dimension is derived for a specific class of GNNs and a generalisation error bound is given using node representations. However, their approach of subsuming the graph convolutions under Pfaffian functions does not allow for an explicit representation in terms of the diffusion operator which is important to our presented analysis. Garg et al. (2020) derives the Rademacher complexity for GNN in a supervised setting with the focus of the equivariant structures of the input graphs and does not allow for an explicit inclusion and analysis of the graph information. Liao et al. (2021) provides PAC-Bayes bounds for GNNs that are tighter than the bounds in Garg et al. (2020).

In the context of this work, especially relevant is Oono et al. (2020b) and Oono et al. (2020a). Oono et al. (2020a) describes the effect of oversmoothing with increasing number of layers. A more detailed comparison to our work is presented in section 2.3. Oono et al. (2020b) analyzes GNNs in the transductive setting. However, they consider a multiscale GCN, and therefore, the analysis is based in a weak-learning/boosting framework where the focus is mostly on exploring the weak learning component, whereas this paper focuses on the specific analysis of the generalization bound and the influence of it's individual components. In addition, we provide a detailed analysis of its dependence on the graph and feature information and provide a more expressive bound by considering generalization under planted models.

**Infinite limit analysis.** In the broader deep learning, there has been a growing call for alternatives to standard learning-theoretic bounds since they do not adequately capture the behaviour of deep models (Neyshabur et al. 2017). To this end, different limiting case analysis have been introduced. In the context of GNNs, it is known that GNNs have a fundamental connection to belief propagation and message passing (Dai et al. 2016; Gilmer et al. 2017) and some theoretical analyses of GNNs have been based on cavity methods and mean field approaches for supervised (Zhou et al. 2020) and transductive settings (Kawamoto et al. 2019; Chen et al. 2019). The central idea of these approaches is to show results in the limit of the number of iterations. In another limiting setting, Du et al. (2019) study GNNs with infintely wide hidden layers, and derive corresponding neural tangent kernel (Jacot et al. 2018; Arora et al. 2019b) that can provide generalisation error bounds in the supervised setting. Keriven et al. (2020) derive continuous versions of GNNs applied to large random graphs. While limiting assumptions allow for a theoretical analysis, it is difficult to infer the implications of these results for finite GNNs.

---

[1] In semi-supervised learning, the learner is given a training set of labeled and unlabeled examples and the goal is to generate a hypothesis that generates predictions on the unseen examples. In transductive learning all features are available to the learner, and the goal is to transfer knowledge from the labeled to the unlabeled data points. The focus of graph-based semi-supervised learning aligns more with the latter setting.

**Contributions and paper structure.** We reconsider classical learning-theoretic measures to analyse GNNs, with a specific focus on explicitly characterising the influence of the graph information and the network architecture on generalisation. In the process, we show that, under careful construction of the complexity measure and distributional assumptions on the graph data, learning theory can provide insights into the behaviour of GNNs. The main contributions are the following:

1) We introduce a formal setup for graph-based transductive inference, and in Section 2.2, we use this framework to show that VC Dimension based generalisation error bounds are typically loose, except for few trivial cases. This observation is along the lines of existing evidence for neural networks.

2) In Section 2.3, we derive generalization bounds based on the transductive Rademacher complexity. Our results show that these bounds are more informative, suggesting that the correct choice of complexity measure is important.

3) We further refine the generalisation error bounds in Section 3 under a planted model for the graph and features. Such an analysis, under random graphs, is rare in GNN literature. We empirically show that the test error is consistent with the trends predicted by the theoretical bound. Our results suggest that, under distributional assumptions, learning-theoretic bounds can explain behaviour of GNNs.

4) In addition, we consider GNNs with residual connections in Section 4, and demonstrate how the above analysis can be extended to other network architectures. We prove that residual connections have smaller generalisation gap in comparison with vanilla GNN, and also empirically show that the theoretical bounds explain (to a limited extent) the influence of network depth on performance.

We conclude in Section 5. All proofs and an overview of the notation are provided in the appendix.

## 2 Statistical Framework for Transductive Learning on GNN

For a rigorous analysis, we introduce a statistical learning framework for graph based transductive inference in Section 2.1. Based on this, we derive generalisation error bounds based on VC Dimension in Section 2.2 and demonstrate that the bounds have limited expresitivity even under strong assumptions. To overcome this problem we consider transductive Rademacher complexity in Section 2.3. While without further assumptions this bound also gives limited insight, the bound is more expressive and, in Section 3, we show that it can provide meaningful bounds under certain distributional assumptions.

### 2.1 Framework for Transductive Learning

We briefly recall the framework for supervised binary classification. Let $\mathcal{X} = \mathbb{R}^d$ be the *domain or feature space* and $\mathcal{Y} = \{\pm 1\}$ be the *label set*. The goal is to find a predictor $h : \mathcal{X} \to \mathcal{Y}$ based on $m$ training samples $S \triangleq \{(\boldsymbol{x}_i, y_i)\}_{i=1}^m \subset \mathcal{X} \times \mathcal{Y}$ and a loss function $\ell : \mathcal{Y} \times \mathcal{Y} \to [0, \infty)$. In a statistical framework, we assume that $S$ consists independent labelled samples from a distribution $\mathcal{D} = \mathcal{D}_{\mathcal{X}} \times \eta$, that is, $\boldsymbol{x}_i \sim \mathcal{D}_{\mathcal{X}}$ and $\boldsymbol{y}_i \sim \eta(\boldsymbol{x}_i)$, where $\eta(\cdot)$ governs the label probability for each feature. The goal of learning is to find $h$ that minimises the *risk / generalisation error* $\mathcal{L}_{\mathcal{D}}(h) \triangleq \mathbb{E}_{(\boldsymbol{x},y)\sim\mathcal{D}}[\ell(h(\boldsymbol{x}), y)]$. Since, $\mathcal{L}_{\mathcal{D}}(h)$ cannot be computed without the knowledge of $\mathcal{D}$, one minimises the *empirical risk* over the training sample $S$ as $\mathcal{L}_S(h) \triangleq \frac{1}{m} \sum_{i=1}^m \ell(h(\boldsymbol{x}_i), y_i)$.

**Transductive learning.** In transductive inference, one restricts the domain to be $\mathcal{X} \triangleq \{\boldsymbol{x}_i\}_{i=1}^n$, a finite set of features $\boldsymbol{x}_i \in \mathbb{R}^d$. Without loss of generality, one may assume that the labels $y_1, \ldots, y_m \in \{\pm 1\}$ are known, and the goal is to predict $y_{m+1}, \ldots y_n$. The problem can be reformulated in the statistical learning framework as follows. We define the feature distribution $\mathcal{D}_{\mathcal{X}}$ to be uniform over the $n$ features, whereas $\boldsymbol{y}_i \sim \eta(\boldsymbol{x}_i)$ for some unknown distribution $\eta$. Hence $\mathcal{D} := \text{Unif}([n]) \times \eta$ is the joint distribution on $\mathcal{X} \times \mathcal{Y}$, and the goal is to find a predictor $h : \mathcal{X} \to \mathcal{Y}$ that minimises the *generalisation error* $\mathcal{L}_u(h) \triangleq \frac{1}{n-m} \sum_{i=m+1}^n \ell(h(x_i), y_i)$. In addition we define the *empirical error* of $h$ to be $\widehat{\mathcal{L}}_m(h) \triangleq \frac{1}{m} \sum_{i=1}^m \ell(h(x_i), y_i)$ and the full sample error of $h$ to be $\mathcal{L}_n(h) \triangleq \frac{1}{n} \sum_{i=1}^n \ell(h(x_i), y_i)$, which is defined over both labelled and unlabelled instances. The purpose of this paper is to derive generalisation error bound for graph based transduction of the form

$$\mathcal{L}_u(h) \leq \widehat{\mathcal{L}}_m(h) + \text{complexity term}.$$

The complexity term is typically characterised using learning-theoretic terms such as VC Dimension and Rademacher complexity. For the transductive setting see Tolstikhin et al. (2016), El-Yaniv et al. (2009), and Tolstikhin et al. (2014).

**Graph-based transductive learning.** A typical view of graph information in transductive inference is as a form of a regularisation (Belkin et al. 2004). In contrast, we view the graph as part of the hypothesis class and derive the impact of the graph information on the complexity term. We assume access to a graph $\mathcal{G}$ with $n$ vertices, corresponding to the respective feature vectors $\boldsymbol{x}_1, \ldots, \boldsymbol{x}_n$, and edge $(i, j)$ denoting similarity of vertices $i$ and $j$. For ease of exposition, we define the matrix $\boldsymbol{X} \in \mathbb{R}^{n \times d}$ with rows being the $n$ feature vectors of dimension $d$. We also abuse notation to write a predictor as $h : \mathbb{R}^{n \times d} \to \{\pm 1\}^n$. Furthermore, typically neural networks output a soft predictor in $\mathbb{R}$, that is further transformed into labels through sign or softmax functions. Hence, much of our analysis focuses on predictors $h : \mathbb{R}^{n \times d} \to \mathbb{R}^n$, and corresponding hypothesis class

$$\mathcal{H}_{\mathcal{G}} = \left\{ h : \mathbb{R}^{n \times d} \to \mathbb{R}^n \ : \ h \text{ is parametrized by } \mathcal{G} \right\} \subset \mathbb{R}^{[n]}.$$

When applicable, we denote the hypothesis class of binary predictors obtained through sign function as $\text{sign} \circ \mathcal{H}_{\mathcal{G}} = \{\text{sign}(h) \mid h \in \mathcal{H}_{\mathcal{G}}\}$. Note that $\text{sign} \circ \mathcal{H}_{\mathcal{G}} \subset \mathcal{H}_{\mathcal{G}}$, and hence, VC Dimension or Rademacher complexity bounds for the latter also hold for the hypothesis class of binary predictors. We also note that the presented analysis holds for both sign and sigmoid function for binarisation.

**Formal setup of GNNs.** We next characterise the hypothesis class for graph neural networks. Consider graph-based neural network model with the propagation rule for layer $k$ denoted by $g_k(\boldsymbol{H}) : \mathbb{R}^{d_{k-1}} \to \mathbb{R}^{d_k}$ with layer wise input matrix $\boldsymbol{H} \in \mathbb{R}^{n \times d_{k-1}}$. Consider a class of GNNs defined over $K$ layers, with dimension of layer $k \in [K]$ being $d_k$ and $\boldsymbol{S} \in \mathbb{R}^{n \times n}$ the graph diffusion operator. Let $\phi$ denote the point-wise activation function of the network, which we assume to be a Lipschitz function with Lipschitz constant $L_\phi$. We assume $\phi$ to be the same throughout the network. We define the hypothesis class over all $K$-layer GNNs as:

$$\mathcal{H}_{\mathcal{G}}^{\phi} \triangleq \left\{ h_{\mathcal{G}}^{\phi}(\boldsymbol{X}) = g_K \circ \cdots \circ g_0 \ : \ \mathbb{R}^{n \times d} \to \{\pm 1\}^n \right\} \tag{1}$$

$$\text{with} \quad g_k \triangleq \phi\left(\boldsymbol{b}_k + \boldsymbol{S} g_{k-1}(\boldsymbol{H}) \boldsymbol{W}_k\right), \ k \in [K], \quad g_0 \triangleq \boldsymbol{X}. \tag{2}$$

where (2) defines the layer wise transformation with $\boldsymbol{W}_k \in \mathbb{R}^{d_{k-1} \times d_k}$ as the trainable weight matrix and $\boldsymbol{b}_k \in \mathbb{R}^{d_k}$ the bias term. Here, the graph is treated as part of the hypothesis class, as indicated by the subscript in $\mathcal{H}_{\mathcal{G}}^{\phi}$. For ease of notation we drop the superscript for non-linearity where it is unambiguous. For the diffusion operator $\boldsymbol{S}$, we consider two main formulations during discussions:

$$\boldsymbol{S}_{\text{loop}} \triangleq \boldsymbol{A} + \mathbb{I} \qquad\qquad\qquad\qquad\qquad\qquad \text{self loop}$$

$$\boldsymbol{S}_{\text{nor}} \triangleq (\boldsymbol{D} + \mathbb{I})^{-\frac{1}{2}} (\boldsymbol{A} + \mathbb{I}) (\boldsymbol{D} + \mathbb{I})^{-\frac{1}{2}}, \qquad \text{degree normalized (Kipf et al. 2017)}$$

where $\boldsymbol{A}$ denotes the graph adjacency matrix and $\boldsymbol{D}$ is the degree matrix. However, most results are stated for general $\boldsymbol{S}$.

## 2.2 Generalisation Error-bound using VC Dimension

The main focus of this paper is the notion of generalisation, that is, understanding how well a GNN can predict the classes of an unlabelled set given the training data. We start with one of the most fundamental learning-theoretical concepts in this context which is the Vapnik–Chervonenkis (VC) dimension of a hypothesis class, a measure of the complexity or expressive power of a space of functions learned by a binary classification algorithm. The following result bounds the VC Dimension for the hypothesis class $\mathcal{H}_{\mathcal{G}}^{\phi}$, and use it to derive a generalisation error bound with respect to the full sample error $\mathcal{L}_n$, which is close to the generalisation error for unlabelled examples $\mathcal{L}_u$ when $m \ll n$.

**Proposition 1 (Generalisation error bound for GNNs using VC Dimension)** *For the hypothesis class over all **linear GNNs**, that is $\phi(x) := x$, with binary outputs, the VC Dimension is given by*

$$\text{VCdim}\left(\text{sign} \circ \mathcal{H}_{\mathcal{G}}^{linear}\right) = \min\left\{ d, \text{rank}(\boldsymbol{S}), \min_{k \in [K-1]} \{d_k\} \right\}.$$

*Similarly, the VC Dimension for the hypothesis class of GNNs with **ReLU non-linearities** and binary outputs, can be bounded as* $\text{VCdim}\left(\text{sign} \circ \mathcal{H}_{\mathcal{G}}^{\text{ReLU}}\right) \leq \min\left\{\text{rank}(\boldsymbol{S}), d_{K-1}\right\}.$

*Using the above bounds, it follows that, for any $\delta \in (0,1)$, the generalisation error for any $h \in$ sign $\circ \mathcal{H}_{\mathcal{G}}$ satisfies, with probability $1 - \delta$,*

$$\mathcal{L}_n(h) - \widehat{\mathcal{L}}_m(h) \leq \sqrt{\frac{8}{m}\left(\min\{\text{rank}(\boldsymbol{S}), d_{K-1}\} \cdot \ln(em) + \ln\left(\frac{4}{\delta}\right)\right)}. \qquad (3)$$

To interpret Proposition 1, we note that, by introducing the non-linearity, we lose the information about the hidden layers, except the last one and therefore also on the feature dimension. Nevertheless, the information on the graph information (that we are primarily interested in) is preserved. There are two situations that arise. If $d_{K-1} \leq \text{rank}(\boldsymbol{S})$, then, from Proposition 1, the graph information is redundant and one could essentially train a fully connected network without diffusion on the labelled features, and use it to predict on unlabelled features. The graph information has an influence for $\text{rank}(\boldsymbol{S}) < d_{K-1}$. While general statements on the influence of the graph information are difficult, by considering specific assumptions on the graph we can characterise the generalisation error further.

For linear GNN on graph $\mathcal{G}$, one can bound the VC Dimension between those for empty and complete graphs, that is, $\text{VCdim}\left(\text{sign} \circ \mathcal{H}_{\text{complete}}^{\text{linear}}\right) \leq \text{VCdim}\left(\text{sign} \circ \mathcal{H}_{\mathcal{G}}^{\text{linear}}\right) \leq \text{VCdim}\left(\text{sign} \circ \mathcal{H}_{\text{empty}}^{\text{linear}}\right)$. Moreover, for disconnected graphs, $\text{rank}(\boldsymbol{S})$ is related to the number of connected components. Similar observations hold for upper bounds on VC Dimension for ReLU GNNs. Based on this observation for simple settings, it holds that considering graph information in comparison to a fully connected feed forward neural network leads to a decrease in the complexity of the class, and therefore also in the generalisation error. However, the graph $\mathcal{G}$ is connected in most practical scenarios, and even under strong assumptions on the graph, for example under consideration of Erdös-Rényi graphs or stochastic block models, $\text{rank}(\boldsymbol{S}) = O(n)$ (Costello et al. 2008). Therefore, for the case $d_{K-1} > \text{rank}(\boldsymbol{S}) = O(n)$, Proposition 1 provides a generalisation error bound of $O\left(\sqrt{\frac{n \cdot \ln m}{m}}\right)$, which holds trivially for 0-1 loss as $n > m$. Furthermore, $\text{rank}(\boldsymbol{S})$ is often similar for both self-loop $\boldsymbol{S}_{\text{loop}}$ and degree-normalised diffusion $\boldsymbol{S}_{\text{nor}}$, and hence, the VC Dimension based error bound does not reflect the positive influence of degree normalisation—a fact that can be explained through stability based analysis (Verma et al. 2019).

### 2.3 Generalisation Error-bound using Transductive Rademacher Complexity

Due to the triviality of VC Dimension based error bounds, we consider generalization error bounds based on transductive Rademacher complexity (TRC). We start by defining TRC that differs from inductive Rademacher complexity by taking the unobserved instances into consideration.

**Definition 1 (Transductive Rademacher complexity (El-Yaniv et al. 2009))** *Let $\mathcal{V} \subseteq \mathbb{R}^n$, $p \in [0, 0.5]$ and $m$ the number of labeled points. Let $\boldsymbol{\sigma} = (\sigma_1, \ldots, \sigma_n)^T$ be a vector of independent and identically distributed random variables, where $\sigma_i$ takes value $+1$ or $-1$, each with probability $p$, and 0 with probability $1 - 2p$. The transductive Rademacher complexity (TRC) of $\mathcal{V}$ is defined as*

$$\mathfrak{R}_{m,n}(\mathcal{V}) \triangleq \left(\frac{1}{m} + \frac{1}{n-m}\right) \cdot \mathbb{E}_{\sigma}\left[\sup_{\mathbf{v} \in \mathcal{V}} \boldsymbol{\sigma}^\top \mathbf{v}\right].$$

The following result derives a bound for the TRC of GNNs, defined in (1)–(2), and states the corresponding generalization error bound. The bound involves standard matrix norms, such as $\|\cdot\|_\infty$ (maximum absolute row sum) and the 'entrywise' norm, $\|\cdot\|_{2\to\infty}$ (maximum 2-norm of any column).

**Theorem 1 (Generalization error bound for GNNs using TRC)** *Consider $\mathcal{H}_{\mathcal{G}}^{\phi,\beta,\omega} \subseteq \mathcal{H}_{\mathcal{G}}^{\phi}$ such that the trainable parameters satisfy $\|\boldsymbol{b}_k\|_1 \leq \beta$ and $\|\boldsymbol{W}_k\|_\infty \leq \omega$ for every $k \in [K]$. The transductive Rademacher complexity (TRC) of the restricted hypothesis class is bounded as*

$$\mathfrak{R}_{m,n}(\mathcal{H}_{\mathcal{G}}^{\phi,\beta,\omega}) \leq \frac{c_1 n^2}{m(n-m)}\left(\sum_{k=0}^{K-1} c_2^k \|\boldsymbol{S}\|_\infty^k\right) + c_3 c_2^K \|\boldsymbol{S}\|_\infty^K \|\boldsymbol{S}\boldsymbol{X}\|_{2\to\infty}\sqrt{\log(n)}, \qquad (4)$$

*where $c_1 \triangleq 2L_\phi\beta$, $c_2 \triangleq 2L_\phi\omega$, $c_3 \triangleq L_\phi\omega\sqrt{2/d}$ and $L_\phi$ is Lipschitz constant for activation $\phi$.*

*The bound on TRC leads to a generalisation error bound following El-Yaniv et al. (2009). For any $\delta \in (0, 1)$, the generalisation error for any $h \in \mathcal{H}_{\mathcal{G}}^{\phi,\beta,\omega}$ satisfies*

$$\mathcal{L}_u(h) - \widehat{\mathcal{L}}_m(h) \leq \mathfrak{R}_{m,n}(\mathcal{H}_{\mathcal{G}}^{\phi,\beta,\omega}) + c_4 \frac{n\sqrt{\min\{m, n-m\}}}{m(n-m)} + c_5\sqrt{\frac{n}{m(n-m)}\ln\left(\frac{1}{\delta}\right)} \quad (5)$$

*with probability $1 - \delta$, where $c_4, c_5$ are absolute constants such that $c_4 < 5.05$ and $c_5 < 0.8$.*

The additional terms in (5) are $O\left(\max\left\{\frac{1}{\sqrt{m}}, \frac{1}{\sqrt{n-m}}\right\}\right)$, and hence, we may focus on the upper bound on TRC (4) to understand the influence of the graph diffusion $\boldsymbol{S}$ as well as its interaction with the feature matrix $\boldsymbol{X}$. The bound depends on the choice of $\omega$, and it suggests a natural choice of $\omega = O(1/\|\boldsymbol{S}\|_\infty)$ such that the bound does not grow exponentially with network depth. The subsequent discussions focus on the dependence on $\|\boldsymbol{S}\|_\infty$ and $\|\boldsymbol{S}\boldsymbol{X}\|_{2\to\infty}$, ignoring the role of $\omega$. Few observations are evident from (4), which are also interesting in comparison to existing works.

**Role of normalisation.** In the case of self-loop, it is easy to see that $\|\boldsymbol{S}_{\text{loop}}\|_\infty = 1 + d_{\max}$, where $d_{\max}$ denotes the maximum degree, and hence, for fixed $\omega$, the bound grows as $O(d_{\max}^K)$. In contrast, for degree normalisation, $\|\boldsymbol{S}_{\text{nor}}\|_\infty = O\left(\sqrt{\frac{d_{\max}}{d_{\min}}}\right)$, and hence, the growth is much smaller (in fact, $\|\boldsymbol{S}_{\text{nor}}\|_\infty = 1$ on regular graphs). It is worth noting that, in the supervised setting, Liao et al. (2021) derived PAC-Bayes for GNN with diffusion $\boldsymbol{S}_{\text{nor}}$, where the bound varies as $O(d_{\max}^K)$. Theorem 1 is tighter in the sense that, for $\boldsymbol{S}_{\text{nor}}$, the error bound has weaker dependence on $d_{\max}$, mainly through $\|\boldsymbol{S}\boldsymbol{X}\|_{2\to\infty}$.

**From spectral radius to $\|\boldsymbol{S}\boldsymbol{X}\|_{2\to\infty}$.** Previous analyses of GNNs in transductive setting rely on the spectral properties of $\boldsymbol{S}$. For instance, the stability based generalisation error bound for 1-layer GNN in Verma et al. (2019) is $O(\|\boldsymbol{S}\|_2^2)$, where $\|\boldsymbol{S}\|_2$ is the spectral norm. In contrast, Theorem 1 shows TRC $= O(\|\boldsymbol{S}\|_\infty \|\boldsymbol{S}\boldsymbol{X}\|_{2\to\infty})$. This is the first result that explicitly uses the relation between the graph-information and the feature information explicitly via $\|\boldsymbol{S}\boldsymbol{X}\|_{2\to\infty}$. One may note that without node features, that is $\boldsymbol{X} = \mathbb{I}$, we have $\|\boldsymbol{S}\|_{2\to\infty} \leq \|\boldsymbol{S}\|_2 \leq \|\boldsymbol{S}\|_\infty$ and hence, a direct comparison between (5) and $O(\|\boldsymbol{S}\|_2^2)$ bound of Verma et al. (2019) is inconclusive. However, in presence of features $\boldsymbol{X}$, Theorem 1 shows that the bound depends on the alignment between the feature and graph information.

In the presence of graph information we can still express Theorem 1 in terms of spectral components by considering $\|\boldsymbol{S}\boldsymbol{X}\|_{2\to\infty} = \max_j \|(\boldsymbol{S}\boldsymbol{X})_{\cdot j}\|_2 \leq \max_j \|\boldsymbol{S}\|_2 \|\boldsymbol{X}_{\cdot j}\|_2 \leq \|\boldsymbol{S}\|_2 \|\boldsymbol{X}\|_{2\to\infty}$ and $\|\boldsymbol{S}\boldsymbol{X}\|_{2\to\infty}$ which can be bound as $\frac{1}{\sqrt{n}}\|\boldsymbol{S}\|_\infty \leq \|\boldsymbol{S}\|_2$.

**Oversmoothing.** While the above bound provides a weaker result than (4) it allows to directly connect to the oversmoothing (Li et al. 2018) effect as the diffusion operator in now only included as $\|\boldsymbol{S}\|_2^k$, $k \in [K]$. Therefore with an increasing number of layers (and especially in the setting considered in Oono et al. (2020a) where the number of layers goes to infinity), the information provided by the graph gets oversmoothed and therefore, a loss of information can be observed.

## 3   Generalization using TRC under Planted Models

The discussion in previous section shows that TRC based generalisation error bound provides some insights into the behaviour of GNNs (example, $\boldsymbol{S}_{\text{nor}}$ is preferred over $\boldsymbol{S}_{\text{loop}}$), but the bound is too general to give insights into the influence of the graph information on the generalisation error. The key quantity of interest is $\|\boldsymbol{S}\boldsymbol{X}\|_{2\to\infty}$, which characterises how the graph and feature information interact. To understand this interaction, we make specific distributional assumptions on both graph and node features. We assume that node features are sampled from a mixture of two $d$-dimensional isotropic Gaussians (Dasgupta 1999), and graph is independently generated from a two-community stochastic block model (Abbe 2018). Both models have been extensively studied in the context of recovering the latent classes from random observations of features matrix $\boldsymbol{X}$ or adjacency matrix $\boldsymbol{A}$, respectively. Our interest, however, is to quantitatively analyse the influence of graph information when the latent classes in features $\boldsymbol{X}$ and graph $\boldsymbol{A}$ do not align completely. In Section 3.1, we present the model and derive bounds on expected TRC, where the expectation is with respect to random features and graph. We then experimentally illustrate the bounds in Section 3.2, and demonstrate that the corresponding generalisation error bounds indeed capture the trends in performance of GNN.

## 3.1 Model and Bounds on TRC

We assume that the node features are sampled latent true classes, given a $\boldsymbol{z} = (z_1, \ldots, z_n) \in \{\pm 1\}^n$. The node features are sampled from a Gaussian mixture model (GMM), that is, feature for node-$i$ is sampled as $\boldsymbol{x}_i \sim \mathcal{N}(z_i \boldsymbol{\mu}, \sigma^2 \mathbb{I})$ for some $\boldsymbol{\mu} \in \mathbb{R}^d$ and $\sigma \in (0, \infty)$. We express this in terms of $\boldsymbol{X}$ as

$$\boldsymbol{X} = \mathcal{X} + \boldsymbol{\epsilon} \in \mathbb{R}^{n \times d}, \qquad \text{where } \mathcal{X} = \boldsymbol{z}\boldsymbol{\mu}^\top \text{ and } \boldsymbol{\epsilon} = (\epsilon_{ij})_{i \in [n], j \in [d]} \overset{i.i.d.}{\sim} \mathcal{N}(0, \sigma^2). \tag{6}$$

We refer to above as $\boldsymbol{X} \sim 2\text{GMM}$. On the other hand, we assume that graph has two latent communities, characterised by $\boldsymbol{y} \in \{\pm 1\}^n$. The graph is generated from a stochastic block model with two classes (2SBM), where edges $(i, j)$ are added independently with probability $p \in (0, 1]$ if $y_i = y_j$, and with probability $q < [0, p)$ if $y_i \neq y_j$. In other words, we define the random adjacency $\boldsymbol{A} \sim 2\text{SBM}$ as a symmetric binary matrix with $\boldsymbol{A}_{ii} = 0$, and $(\boldsymbol{A}_{ij})_{i<j}$ indenpendent such that

$$\boldsymbol{A}_{ij} \sim \text{Bernoulli}(\mathcal{A}_{ij}), \qquad \text{where } \mathcal{A} = \frac{p+q}{2}\mathbf{1}\mathbf{1}^\top + \frac{p-q}{2}\boldsymbol{y}\boldsymbol{y}^\top - p\mathbb{I}. \tag{7}$$

The choice of two different latent classes $\boldsymbol{z}, \boldsymbol{y} \in \{\pm 1\}^n$ allows study of the case where the graph and feature information of do not align completely. We use $\Gamma = |\boldsymbol{y}^\top \boldsymbol{z}| \in [0, n]$ to quantify this alignment. Assuming $\boldsymbol{y}, \boldsymbol{z}$ are both balanced, that is, $\sum_i y_i = \sum_i z_i = 0$, one can verify that

$$\|(\mathcal{A} + \mathbb{I})\mathcal{X}\|_{2\to\infty} = \|\boldsymbol{\mu}\|_\infty \left(n(1-p)^2 + \tfrac{1}{4}n(p-q)^2\Gamma^2 - (p-q)(1-p)\Gamma^2\right)^{1/2}, \tag{8}$$

which indicates that, for dense graphs ($p, q \gg \frac{1}{n}$), the quantity $\|\boldsymbol{SX}\|_{2\to\infty}$ should typically increase if the latent structure of graph and features are more aligned. This intuition is made precise in the following result that bounds the TRC, in expectation, assuming $\boldsymbol{X} \sim 2\text{GMM}$ and $\boldsymbol{A} \sim 2\text{SBM}$.

**Theorem 2 (Expected TRC for GNNs under SBM)** *Let $c_1, c_2$ and $c_3$ as defined in Theorem 1 and $\Gamma \triangleq |\boldsymbol{y}^\top \boldsymbol{z}|$. Let $c_6 \triangleq (1 + o(1))$, $c_7 \triangleq (1 + ko(1))$, $c_8 \triangleq (1 + Ko(1))$. Assuming $p, q \gg \frac{(lnn)^2}{n}$ we can bound the expected TRC for $\boldsymbol{A}$ as defined in* (7) *and $\boldsymbol{X}$ as defined in* (6) *as follows:*

*Case 1, Degree normalized: $\boldsymbol{S} = \boldsymbol{S}_{nor}$*

$$\mathbb{E}_{\substack{\boldsymbol{X}\sim 2\text{GMM} \\ \boldsymbol{A}\sim 2\text{SBM}}} \left[\mathfrak{R}_{m,n}(\mathcal{H}_\mathcal{G}^{\phi,\beta,\omega})\right] \leq \frac{c_1 n^2}{m(n-m)} \left(\sum_{k=0}^{K-1} c_7 c_2^k \left(\frac{p}{q}\right)^{\frac{k}{2}}\right) + c_8 c_3 c_2^K \left(\frac{p}{q}\right)^{\frac{K}{2}} \sqrt{\ln(n)} \times$$
$$\left(c_6 \|\boldsymbol{\mu}\|_\infty \frac{1 + \left(\frac{p-q}{2}\right)^2 \Gamma^2}{\left(\frac{p+q}{2}\right)^2} + c_6 \sqrt{\frac{\ln(n)}{q}} \|\boldsymbol{\mu}\|_\infty + c_6 \sqrt{\frac{\sigma(1 + 2\ln(d))}{q}}\right) \tag{9}$$

*Case 2, Self Loop: $\boldsymbol{S} = \boldsymbol{S}_{loop}$*

$$\mathbb{E}_{\substack{\boldsymbol{X}\sim 2\text{GMM} \\ \boldsymbol{A}\sim 2\text{SBM}}} \left[\mathfrak{R}_{m,n}(\mathcal{H}_\mathcal{G}^{\phi,\beta,\omega})\right] \leq \frac{c_1 n^2}{m(n-m)} \left(\sum_{k=0}^{K-1} c_7 c_2^k (np)^k\right) + c_8 c_3 c_2^K (np)^K \sqrt{\ln(n)} \times$$
$$\left(c_6 \|\boldsymbol{\mu}\|_\infty n\left(1 + \left(\frac{p-q}{2}\right)^2 \Gamma^2\right) + n\sqrt{\frac{p+q}{2}} \|\boldsymbol{\mu}\|_\infty + c_6 n\sqrt{p}\sigma\sqrt{1 + 2\ln(d)}\right) \tag{10}$$

We note that although the above bounds are stated in expectation, they can be translated into high probability bounds. Furthermore the non-triviality of the proof of Theorem 2 stems from bounds on the expectations of matrix norms, which is more complex than the computation in (8). Theorem 2 can be also translated into bounds on the generalisation gap $\mathcal{L}_u(h) - \widehat{\mathcal{L}}_m(h)$. By considering a planted model we can now further extend the observations of Section 2.2 and 2.3.

**Role of normalisation.** In the following, we can show that by normalising, the generalisation gap grows slower with increasing graph size. First we compare $\mathbb{E}\left[\|\boldsymbol{S}_{\text{loop}}\|_\infty^k\right] = c_7(np)^k$ with $\mathbb{E}\left[\|\boldsymbol{S}_{\text{nor}}\|_\infty^k\right] = c_7 (p/q)^{k/2}$ and observe that by normalising we lose the $n$ term. In addition we can consider $\mathbb{E}\left[\|\boldsymbol{SX}\|_{2\to\infty}\right]$ which is bound by the second line in (9)–(10). Again in the first, deterministic, term we observe that the self loop version contains an additional dependency on $n$. For the two noise terms we can characterize the behaviour in terms of the density of the graph. Let

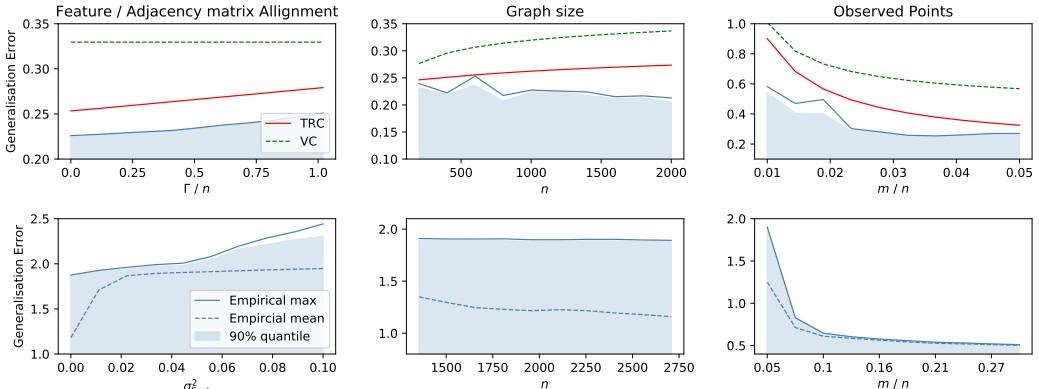

Figure 1: *Top row* shows experiments for SBM and *bottom row* for Cora. *(left)* Change in the alignment of the features and adjacency matrix. *(middle)* Change of the graph size $n$. *(right)* Change number of observed points $m$.

$\rho = O(p), O(q)$ and $\rho \gg \frac{1}{n}$ then we can characterise the *dense setting* as $\rho \asymp \Omega(1)$ and the *sparse setting* as $\rho \asymp O\left(\frac{\ln(n)}{n}\right)$ and observe that in both case the normalised case grows slower with $n$:

Dense: $\quad \mathbb{E}\left[\|\boldsymbol{S}_{\text{loop}}\boldsymbol{X}\|_{2\to\infty}\right] = O(n)$ $\qquad$ and $\quad \mathbb{E}\left[\|\boldsymbol{S}_{\text{nor}}\boldsymbol{X}\|_{2\to\infty}\right] = O(\sqrt{\ln(n)})$ $\quad$ (11)

Sparse: $\quad \mathbb{E}\left[\|\boldsymbol{S}_{\text{loop}}\boldsymbol{X}\|_{2\to\infty}\right] = O(\sqrt{n\ln(n)})$ $\quad$ and $\quad \mathbb{E}\left[\|\boldsymbol{S}_{\text{nor}}\boldsymbol{X}\|_{2\to\infty}\right] = O(\sqrt{n})$ $\quad$ (12)

**Influence of the graph information.** We consider the idea from Section 2.2, to analyse the influence of graph information by comparing the TRC between the case where no graph information is considered, $\boldsymbol{S} = \mathbb{I}$ and $\boldsymbol{S}_{\text{nor}}$. We define the corresponding hypothesis classes as $\mathcal{H}_{\mathbb{I}}^{\phi,\beta,\omega}$ and $\mathcal{H}_{\text{nor}}^{\phi,\beta,\omega}$. Considering the deterministic case ($\boldsymbol{S} = \mathcal{S}, \boldsymbol{X} = \mathcal{X}$) we can observe $\mathfrak{R}_{m,n}(\mathcal{H}_{\mathbb{I}}^{\phi,\beta,\omega}) > \mathfrak{R}_{m,n}(\mathcal{H}_{\text{nor}}^{\phi,\beta,\omega})$ if $\Gamma > O\left(\frac{n}{\sqrt{n\rho+n}}\right)$. Therefore the random graph setting allows us to more precisely characterize under what conditions adding graph information helps.

## 3.2 Experimental Results

While we focus on the theoretical analysis of GNNs, in this section we illustrate that the empirical generalization error follows the trends given by the bounds described in Theorem 2. The bounds in Section 3.1 are derived for binary SBMs so we therefore focus on this setting but in addition also show that those observations extend to real world, multi-class data on the example of the Cora dataset (Rossi et al. 2015). The results are presented in Figure 1. For the SBM we consider a graph with $n = 500, m = 100$ as default. We plot the mean over 5 random initialisation and over several epochs. Note that the range for Cora exceeds $(0,1)$ as the dataset is multi class and we consider a negative log likelihood loss. For plotting the theoretical bound we can only plot the trend of the bound as the absolute value is out of the $(0,1)$ range. While this does not allow us to numerically show how tight the bound is in practice, we can still make statements about the influence of the change of parameters, where the experiments validate the constancy between theory and empirical observations[2]. Details on the experimental setup are given in the Appendix.

We can first look at the *feature and graph alignment* as characterised through $\Gamma^2$ in the TRC based bound (8)–(9) and observe that with an increase in the latent structure the generalisation error

---

[2]Generalisation error bounds, even for simple machine learning models, can exceed 1 due to absolute constants that cannot be precisely estimated. Hence, the point of interest is the dependence of key parameters; for instance, in a supervised setting, the bounds are $O(1/\sqrt{m})$ and typically exceeds 1 for moderate $m$. This problem is inherent to the bound given in El-Yaniv et al. (2009) that we base our TRC bounds on, as the slack terms can already exceeds 1 and therefore further research on general TRC generalisation gaps is necessary to characterise the absolute gap between theory and experiments.

increases. While this seems to be counterintuitive a possible explanation could be that reduced alignment helps to prevent overfitting and we observe that the slope matches the empirical results. In addition we note that the VC dimension bound (3) does not allow us to model this dependency. For Cora we do not have access to the ground truth for the alignment and therefore can not verify this trend directly. Therefore we simulate a change in the feature structure by adding noise to the feature vector as $\boldsymbol{X} + \epsilon$ where $\epsilon_i$ is $i.i.d.$ distributed $\mathcal{N}(0, \sigma_{\text{Feat}}^2 \mathbb{I})$ and again observe a similar behaviour to the SBM. To be able to apply the bound to arbitrary graphs an important property is that the bound does not increase drastically with growing *graph size*. We theoretically showed this in the previous section, especially through (11)–(12) and illustrate it in Figure 1 (middle). Empirically for both, SBM and Cora, the generalisation error stays mostly consistent over varying $n$. Finally for the *number of observed points* we consider a realistic setting of $m \ll n - m$ where we see a sharp decline in the setting of few observed points but then the generalisation error converges which corresponds to the influence of $m$ as described in (9). Practically such an observation can be useful as labeling data can be expensive and such results could be useful to determine a necessary and sufficient number of labeled data to obtain a given level of accuracy.

# 4  Influence of Depth and Residual Connections on the Generalisation Error

While for standard neural networks increasing the depth is a common approach for increasing the performance, this idea becomes more complex in the context of GNNs as each layer contains a left multiplication of the diffusion operator and we can therefore observe an over-smoothing effect (Li et al. 2018) — the repeated multiplication of the diffusion operator in each layer spreads the feature information such that it converges to be constant over all nodes. To overcome this problem, empirical works suggest the use of residual connections (Kipf et al. 2017; Chen et al. 2020), such that by adding connections from previous layers the network retains some feature information. In this section we investigate this approach in the TRC setting. In Section 4.1 we provide the TRC bound for GNN with skip connections and show that it improves the generalisation error compared to vanilla GNNs. In Section 4.2 we illustrate this bounds empirically.

## 4.1  Model and bounds on TRC for GNN with Residual connections

While there is a wide range of residual connections, introduced in recent years we follow the idea presented in Chen et al. (2020) where a GNN as defined in (2) is extended by an interpolation over parameter $\alpha$ with the features. This setup is especially interesting as it captures the idea of preserving the influence of the feature information more than residual definition that only connect to the previous layer. Formally we can now write the layer wise propagation rule as

$$g_{k+1} \triangleq \phi\left((1 - \alpha)\left(\boldsymbol{b}_k + \boldsymbol{S}g_k\left(\boldsymbol{H}\right)\boldsymbol{W}_k\right) + \alpha g_0\left(\boldsymbol{H}\right)\right), \qquad \text{with } \alpha \in (0, 1). \tag{13}$$

We can now derive a generalization error bound similar to Theorem 1 for the Residual network.

**Theorem 3 (TRC for Residual GNNs)** *Consider a Residual network as defined in* (13) *and* $\mathcal{H}_{\mathcal{G}}^{\phi,\beta,\omega} \subset \mathcal{H}_{\mathcal{G}}^{\phi}$ *such that the trainable parameters satisfy* $\|\boldsymbol{b}_k\|_1 \leq \beta$ *and* $\|\boldsymbol{W}_k\|_\infty \leq \omega$ *for every* $k \in [K]$. *Then with* $\alpha \in (0, 1)$ *and* $c_1 \triangleq 2L_\phi\beta$, $c_2 \triangleq 2L_\phi\omega$, $c_3 \triangleq L_\phi\omega\sqrt{2/d}$ *the TRC of the restricted class or Residual GNNs is bounded as*

$$\mathfrak{R}_{m,n}(\mathcal{H}_{\mathcal{G}}^{\phi,\beta,\omega}) \leq \frac{((1-\alpha)c_1 + \alpha 2L_\phi \|\boldsymbol{X}\|_\infty)n^2}{m(n-m)} \left(\sum_{k=0}^{K-1}(1-\alpha)c_2^k \|\boldsymbol{S}\|_\infty^k\right)$$
$$+ \alpha 2L_\phi \|\boldsymbol{X}\|_\infty + (1-\alpha)c_3 c_2^K \|\boldsymbol{S}\|_\infty^K \|\boldsymbol{S}\boldsymbol{X}\|_{2\to\infty} \sqrt{\log(n)} \tag{14}$$

However observing the bound isolated does not provide new insights beyond Theorem 2 into the behaviour of the generalisation error and therefore we focus on the comparison between GNNs with and without residual connections.

For readability assume $\beta = \|\boldsymbol{X}\|_\infty$. Under this setup we can note that the generalisation error-bound increases with decreased alpha and in extension it follows that the generalisation error-bound for a GNN with skip connection is lower then the one without. This implication is in line with the general notion that residual connections improve the performance of networks (Chen et al. 2020; Kipf et al. 2017). Our general intuition behind this behavior is that with increasing $\alpha$, the network architecture

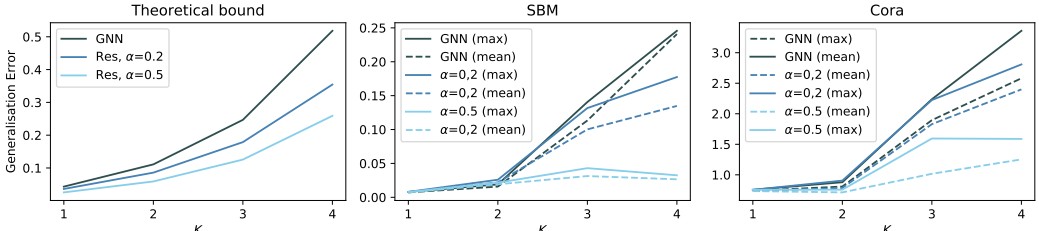

Figure 2: *(left)* Theoretical bounds corresponding to Theorem 3. *(middle)* Influence of depth $K$ under SBM. *(right)* Influence of depth $K$ for *Cora*.

is closer to the one of an one hidden layer network. Having good performance in shallow networks is something that is observed in our experiments as well as in previous work (e.g., (Kipf et al. 2017)). Therefore it appears that using the skip connection to obtain a deep network that resembles a shallow one leads to the performance increase.

## 4.2  Experiments on depth and Residual networks

The above observation suggests that including residual connections is beneficial with increasing depth which is consistent with the initial reason of introducing residual connections (Chen et al. 2020; Kipf et al. 2017). We further illustrate this in the context of the trend in (14). Similar to Section 3.2 we start by considering the vanilla GNN version and focus on the *influence of depth* where Figure 2 (left) illustrates Theorem 2, more specifically an exponential increase of $K$ as shown in (9)–(10) (similar to Liao et al. (2021)). Empirically from Figure 2, (middle, right) we note that with increasing depth the generalisation error indeed increases for the first three layers significantly but then we observe a deviation from the theoretical bound. The rate of growth decreases, which is to be expected as the absolute values of $\mathcal{L}_u, \mathcal{L}_m$ are bound by construction. Future work with a focus on depth is necessary to refine this component of the bound. Extending the analysis of depth we now consider the *residual connections* as defined in (13). By (14) we can still observe the exponential dependency on $K$ and therefore focus on two main aspects: i) Theoretically the generalisation error for the Resnet is upper bound by GNN, which empirically is observed for both the SBM as well as for Cora. ii) Focusing on the Resnets, Theorem 3 predicts an ordering in the generalisation error given by $\alpha$ which is again observed for both the SBM as well as for Cora. Therefore while there seems to be deviation in the exponential behaviour of $K$ as given in Theorem 3, the ordering of the generalisation error-bound described by $\alpha$ is observed empirically. While this does not give us a complete picture we can note that the remarks on oversmoothing suggest that shallower networks are preferable and we again note that the VC dimension bound (3) does not provide any useful insights to the influence of depth.

## 5  Conclusion

Statistical learning theory has proven to be a successful tool for a complete and rigours analysis of learning algorithms. At the same time research suggests that applied to deep learning models these methods become non-informative. However on the example of GNNs, we demonstrate that classical statistical learning theory can be used under consideration of the right complexity measure and distributional assumptions on the data to provide insight into trends of deep models. Our analysis provides first fundamental results on the influence of different parameters on generalization and opens up different lines of follow up work. As noted in the previous section the TRC bound predicts an exponential dependency on network depth $K$ which can only partially be observed empirically and therefore a study without relying on a recursive proof structure will be necessary to refine this dependency on $K$. As it is not the focus of this paper we consider the bounds on the norms of trainable parameters, $\omega, \beta$, fixed. However loosening this assumption would allow us to analyse the behaviour of the generalisation error during training and under different optimization approaches. Considering the current setup we can also extend the theoretical analysis to more advanced architectures such as dropout or batch normalisation. Finally while our analysis focuses on generalisation we suggest that the idea of analysing GNNs under planted models can be extended to other learning-theoretical measures such as stability or model selection as well as the supervised (graph-classification) setting.

# 6 Acknowledgement

This work has been supported by the German Research Foundation (Priority Program SPP-2298, project GH-257/2-1) and the Baden-Württemberg Stiftung (Eliteprogram for Postdocs project "Clustering large evolving networks"). The authors thank the International Max Planck Research School for Intelligent Systems (IMPRS-IS) for supporting Leena Chennuru Vankadara.

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
