# Appendix

In the appendix we provide the following additional information and proofs.

# A Notation

Let $[n] := 1, 2, \ldots n$. We represent a graph $\mathcal{G}$ by its adjacency matrix $\boldsymbol{A}$, and use $\mathbb{I}$ to denote an identity matrix. For any vertex $i$, $i \sim j := \{j \mid \boldsymbol{A}_{ij} = 1\}$ is the set indices adjacent to $i$. We use $\| \cdot \|_p$ to denote the $p$-norm for vectors and induced $p$-norm for matrices. We consider standard matrix norms, such as $\| \cdot \|_\infty$ (maximum absolute row sum) and the 'entrywise' norm, $\| \cdot \|_{2\to\infty}$ (maximum 2-norm of any column). Function classes are denoted as $\mathcal{H}$ or $\mathcal{F}$, indexed depending on parameters that are included in the hypothesis class. We define the fully connected graph as $\mathcal{G} =: K_{\mathcal{G}}$, the empty graph (without any edges) as $\mathcal{G} =: \emptyset$. Note that if we consider a graph with only self loops ($\mathcal{G} := \emptyset$) the GNN becomes equivalent to a fully connected neural network. We consider point wise activation functions $\phi(\cdot) : \mathbb{R} \to \mathbb{R}$. In this context we define the rectified linear unit as $\mathrm{ReLU}(x) := \max\{0, x\}$.

# B  Proof Proposition 1 — Generalisation error bound for GNNs using VC-Dimension

**Definition 2 (VC-Dimension)** *Following Vapnik et al. (1971). Let $\mathcal{H} \subseteq \{\pm 1\}^{\mathcal{X}}$ be a binary function class and $h \in \mathcal{H}$ a function in this class. We define $C = (x_1, \cdots x_m) \in \mathcal{X}^m$ and say that $C$ is shattered by $h$ if for all assignments of labels to points in $C$ there exists a parameterization of $h$ such that $h$ predicts all points in $C$ without error. From there we define the **VC-dimension** of a non-empty hypothesis class $\mathcal{H}$ as the cardinality of the largest possible subset of $\mathcal{X}$ that can be shattered by $\mathcal{H}$. If $\mathcal{H}$ can shatter arbitrarily large sets, then $\mathrm{VCdim}(\mathcal{H}) = \infty$.*

## B.1  Generalization using VC-Dimension under specific graph assumptions

For the hypothesis class over all **linear GNNs**, that is $\phi(x) := x$, with binary outputs, the VC Dimension is given by

$$\mathrm{VCdim}\big(\mathrm{sign} \circ \mathcal{H}_{\mathcal{G}}^{\mathrm{linear}}\big) = \min\big\{d, \mathrm{rank}(\boldsymbol{S}), \min_{k \in [K-1]}\{d_k\}\big\}.$$

Similarly, the VC Dimension for the hypothesis class of GNNs with **ReLU non-linearities** and binary outputs, can be bounded as $\mathrm{VCdim}\big(\mathrm{sign} \circ \mathcal{H}_{\mathcal{G}}^{\mathrm{ReLU}}\big) \leq \min\{\mathrm{rank}(\boldsymbol{S}), d_{K-1}\}$.

Using the above bounds, it follows that, for any $\delta \in (0,1)$, the generalisation error for any $h \in \mathrm{sign} \circ \mathcal{H}_{\mathcal{G}}$ satisfies, with probability $1 - \delta$,

$$\mathcal{L}_n(h) - \widehat{\mathcal{L}}_m(h) \leq \sqrt{\frac{8}{m}\left(\min\{\mathrm{rank}(\boldsymbol{S}), d_{K-1}\} \cdot \ln(em) + \ln\left(\frac{4}{\delta}\right)\right)}.$$

*Proof.* For this proof we will need the following know result on the VC-dimension of linearly independent points:

**Theorem 4 (Burges (1998))** *Consider some set of m points in $\mathbb{R}^n$. Choose any one of the points as origin. Then the $m$ points can be shattered by oriented hyperplanes if and only if the position vectors of the remaining points are linearly independent.*

For deriving $\mathrm{VCdim}\big(\mathrm{sign} \circ \mathcal{H}_{\mathcal{G}}^{\mathrm{linear}}\big)$ we start with the VC-dimension of the final layer: $\mathrm{VCdim}(\mathcal{H}_{\boldsymbol{B}}^{\mathrm{sign}})$ with

$$\mathcal{H}_{\boldsymbol{B}}^{\mathrm{sign}} = \big\{h_{\boldsymbol{B}}^{\mathrm{sign}}(x) := \mathrm{sign}(\boldsymbol{B}\boldsymbol{w}) \; : \; \boldsymbol{w} \in \mathbb{R}^m\big\}$$

over an arbitrary matrix $\boldsymbol{B} \in \mathbb{R}^{n \times m}$, where $\boldsymbol{B}$ is later substituted for the linear network. Let $\mathrm{rank}(\boldsymbol{B}) = r$ then we show that there is $c \subset [n], |c| = r$ s.t. $\forall b \in \{\pm 1\}^r$ and $h_{\boldsymbol{B}}^{\mathrm{sign}}(c) = \{\pm 1\}^c$. Using SVD we decompose $\boldsymbol{B} = \boldsymbol{U}\boldsymbol{\Lambda}\boldsymbol{V}^{\top}$ and define $\boldsymbol{z}_1^{\top}, \cdots, \boldsymbol{z}_m^{\top} \in \mathbb{R}^k$ as the rows of $\boldsymbol{U}$. Using this we rewrite:

$$\boldsymbol{B}\boldsymbol{w} = \begin{bmatrix} \boldsymbol{z}_1^{\top} \\ \vdots \\ \boldsymbol{z}_d^{\top} \end{bmatrix} \underbrace{\boldsymbol{\Lambda}\boldsymbol{V}^{\top}\boldsymbol{w}}_{=\boldsymbol{a} \in \mathbb{R}^d} = \begin{bmatrix} \boldsymbol{z}_1^{\top}\boldsymbol{a} \\ \vdots \\ \boldsymbol{z}_d^{\top}\boldsymbol{a} \end{bmatrix}$$

Rewrite $\mathcal{H}_{\boldsymbol{B}}^{\mathrm{sign}}$ as $\mathcal{F}^{\mathrm{sign}} = \big\{h_a(z) = \mathrm{sign}(\boldsymbol{a}^{\top}\boldsymbol{z})\big\}$. Since $\mathcal{F}^{\mathrm{sign}}$ lies in the class of all homogenious linear classifiers in $r$ dimensions and from orthonormal condition on $\boldsymbol{z}$ it follows that $\mathrm{span}(\{\boldsymbol{z}_1, \cdots \boldsymbol{z}_n\}) = \mathbb{R}^r$. Using this observation as well as results on the VC-dimension of linear independent pointsets Burges (1998) it follows that $\mathrm{VCdim}(\mathcal{H}_{\boldsymbol{B}}^{\mathrm{sign}}) = \mathrm{VCdim}(\mathcal{F}^{\mathrm{sign}}) = r$. Substituting $\boldsymbol{B}$ with the linear network and using that for two matrixes $\boldsymbol{B}'$ and $\boldsymbol{B}$: $\mathrm{rank}(\boldsymbol{B}'\boldsymbol{B}) = \min(\mathrm{rank}(\boldsymbol{B}'), \mathrm{rank}(\boldsymbol{B}))$ gives

$$\mathrm{rank}(\boldsymbol{B}) := \mathrm{rank}(\boldsymbol{S}\boldsymbol{H}^{(p)}) = \mathrm{rank}(\boldsymbol{S} \cdots \boldsymbol{S}\boldsymbol{X}\boldsymbol{W}^{(1)} \cdots \boldsymbol{W}^{(p-1)})$$

as the final result.

For extending to the non-linear setting we first note that we can not make a general statement on the rank of a matrix after applying a non-linearity. That is for some matrix $\boldsymbol{M}$ and non-linearity $\mathrm{ReLU}(\cdot)$ we have no order relation between $\mathrm{rank}(\boldsymbol{M})$ and $\mathrm{rank}(\mathrm{ReLU}(\boldsymbol{M}))$. This can be checked

by a simple counterexample. Therefore the above presented proof does not extend to the hidden layer size but since the last layer is linear the dependency on $S$ persists. We define the hypothesis class over all linear GNNs where all but the last activation function are linear $\phi_k(x) := x \ \forall k \in [K-1]$ and $\phi_p(x) := \text{sign}(x)$ as:

$$\mathcal{H}_{\mathcal{G}}^{\text{sign},\mathbb{I}} = \left\{ h_{\mathcal{G}}^{\text{sign},\mathbb{I}}(\boldsymbol{X}) \right\}$$

and recall that layer $k$ has dimension $d_k$. Then the VC-Dimension is given by the minimum of the rank of the adjacency matrix, the dimension of the features and the minimum hidden layer size, that is,

$$\text{VCdim}\left(\mathcal{H}_{\mathcal{G}}^{\text{sign},\mathbb{I}}\right) = \min\left\{d, \text{rank}(\boldsymbol{S}), \min_{k \in [K-1]}\{d_k\}\right\}. \tag{15}$$

Therefore consider the hypothesis class GNNs with of non-linearities $\phi_k(x) := \text{ReLU}(x) \ \forall k \in [K-1]$ and $\phi_p(x) := \text{sign}(x)$:

$$\mathcal{H}_{\mathcal{G}}^{\text{sign},\text{ReLU}} = \left\{ h_{\mathcal{G}}^{\text{sign},\text{ReLU}}(\boldsymbol{X}) \right\}$$

and again compute the VC-Dimension, similar to the proof shown above, we can note that we lose information on the hidden layers (and therefore also on $d$) and the bound becomes

$$\text{VCdim}\left(\mathcal{H}_{\mathcal{G}}^{\text{sign},\text{ReLU}}\right) \le \min\left\{\text{rank}(\boldsymbol{S}), d_{p-1}\right\}, \tag{16}$$

that is, it still depends on the rank of $S$ but only on the last hidden layer dimension.

Following defined we use the a standard result for generalisation e.g. in Shalev-Shwartz et al. (2014). For $\delta \in (0,1)$ any $h \in \mathcal{H}_{\mathcal{G}}$ satisfies

$$\mathcal{L}_n(h) - \widehat{\mathcal{L}}_m(h) \le \sqrt{\frac{8\left(\text{VCdim}\left(\mathcal{H}_{\mathcal{G}}\right)\ln\left(\frac{em}{\text{VCdim}(\mathcal{H}_{\mathcal{G}})}\right) + \ln\left(\frac{4}{\delta}\right)\right)}{m}} \tag{17}$$

with probability $1-\delta$.

Applying (15) and (16) to (17) gives the final bound. $\qquad\square$

## B.2 Additional notes on the remarks related to Proposition 1

**Expected Rank of Erdös-Rényi graphs** From Costello et al. (2008) we know the following result: Let $c$ be a constant larger then $\frac{1}{2}$, then for any $\frac{c \ln n}{n} \le p \le \frac{1}{2}$ for a random $\mathcal{G}$ graph sampled from a Erdös-Rényi graph has $\text{rank}(\boldsymbol{A}) \le n - i(\mathcal{G})$ with probability $1 - \mathcal{O}\left((\ln \ln n)^{-\frac{1}{4}}\right)$, where $i(\mathcal{G})$ denotes the number of isolated vertices in $\mathcal{G}$.

In the same line we can additionally note that we get similar results (of the form that in expectation $\text{rank}(\boldsymbol{A}) = n$) for more complex models like stochastic block models which we will discuss in further detail later, as for any matrix $\boldsymbol{A} \in \mathbb{R}^{n \times n}$ there are invertible matrices arbitrarily close to $\boldsymbol{A}$, under any norm for the $n \times n$ matrices. Motivated by those first findings we consider a different complexity measure, less reliant on combinatorial arguments, to get more insight into the role of graph information.

# C Proof Theorem 1 — Generalization error bound for GNNs using TRC

Recall the definition of TRC as defined in El-Yaniv et al. (2009)[3]: Let $\mathcal{V} \subseteq \mathbb{R}^n$, $p \in [0, 0.5]$ and $m$ the number of labeled points. Let $\boldsymbol{\sigma} = (\sigma_1, \ldots, \sigma_n)^T$ be a vector of independent and identically distributed random variables, where $\sigma_i$ takes value $+1$ or $-1$, each with probability $p$, and $0$ with probability $1 - 2p$. The transductive Rademacher complexity (TRC) of $\mathcal{V}$ is defined as

$$\mathfrak{R}_{m,n}(\mathcal{V}) \triangleq \left( \frac{1}{m} + \frac{1}{n-m} \right) \cdot \underset{\sigma}{\mathbb{E}} \left[ \sup_{\mathbf{v} \in \mathcal{V}} \boldsymbol{\sigma}^\top \mathbf{v} \right].$$

For this section we introduce the following notation: $Q \triangleq \left( \frac{1}{m} + \frac{1}{n-m} \right)$, which we later again substitute in the final expression.

To derive the TRC we start with the following propositions describing the recursive TRC for a GNN neuron that is applied $K - 1$ times for all but the first layer.

**Proposition 2 (Recursive TRC of one GNN neuron)** *Consider* $g_{k+1} \triangleq \phi \left( \boldsymbol{b}_k + \boldsymbol{S} g_k \left( \boldsymbol{H} \right) \boldsymbol{W}_k \right)$, $k \in \{1, \cdots, K\}$. *Now we define the function class over one neuron as*

$$\mathcal{H}_{\mathcal{G}}^\phi \triangleq \left\{ h_{\mathcal{G}}^\phi(\boldsymbol{H}) = \phi \left( \boldsymbol{b}_i + \sum_l^{d_k} \boldsymbol{W}_{lj} \sum_t^n \boldsymbol{S}_{it} g(\boldsymbol{H})_{lj} \right) \;\middle|\; g \in \mathcal{F}, \|\boldsymbol{b}_i\|_1 \le \beta \right\}$$

*where* $\mathcal{F}$ *is the class of* $\mathbb{R}^{n \times d_k} \to \mathbb{R}$, *including the zero function. Then with* $\boldsymbol{W}_{\cdot j} \triangleq [\boldsymbol{W}_{1j}, \cdots, \boldsymbol{W}_{d_k j}]^\top$:

$$\mathfrak{R}_{m,n}(\mathcal{H}_{\mathcal{G}}^\phi) \le 2 L_\phi \left( \beta Q(n) + \|\boldsymbol{S}\|_\infty \|\boldsymbol{W}_{\cdot j}\|_1 \mathfrak{R}_{m,n}(\mathcal{F}) \right)$$

*Proof.* See section C.2 □

After the recursive application we end up with a formulation of all layers and a dependency on the TRC of the first layer. Therefore we then use the following proposition to finish the proof.

**Proposition 3 (Bound on TRC, first layer)** *Define the hypothesis class over the function of the first layer* $g_0$ *as:*

$$\mathcal{H}_{\mathcal{G}}^\phi \triangleq \left\{ h_{\mathcal{G}}^\phi(\boldsymbol{X}) = \phi \left( \boldsymbol{b} + \boldsymbol{S} \boldsymbol{X} \boldsymbol{W}_1 \right) \;\middle|\; \|\boldsymbol{b}\|_1 \le \beta \right\}$$

*then the TRC is give by*

$$\mathfrak{R}_{m,n}(\mathcal{H}_{\mathcal{G}}^\phi) \le L_\phi \left( \beta Q(n) 2 + Q \|\boldsymbol{W}_1\|_\infty \|\boldsymbol{S} \boldsymbol{X}\|_{2 \to \infty} \sqrt{\frac{2 \log(n)}{d}} \right)$$

*Proof.* See section C.3 □

Then by combining the above results we obtain Theorem 1 as follows: Consider $\mathcal{H}_{\mathcal{G}}^{\phi,\beta,\omega} \subseteq \mathcal{H}_{\mathcal{G}}^\phi$ such that the trainable parameters satisfy $\|\boldsymbol{b}_k\|_1 \le \beta$ and $\|\boldsymbol{W}_k\|_\infty \le \omega$ for every $k \in [K]$. The transductive Randemacher complexity (TRC) of the restricted hypothesis class is bounded as

$$\mathfrak{R}_{m,n}(\mathcal{H}_{\mathcal{G}}^{\phi,\beta,\omega}) \le \frac{c_1 n^2}{m(n-m)} \left( \sum_{k=0}^{K-1} c_2^k \|\boldsymbol{S}\|_\infty^k \right) + c_3 c_2^K \|\boldsymbol{S}\|_\infty^K \|\boldsymbol{S} \boldsymbol{X}\|_{2 \to \infty} \sqrt{\log(n)},$$

where $c_1 \triangleq 2 L_\phi \beta$, $c_2 \triangleq 2 L_\phi \omega$, $c_3 \triangleq L_\phi \omega \sqrt{2/d}$ and $L_\phi$ is Lipschitz constant for activation $\phi$.

---

[3]Note that El-Yaniv et al. (2009) considered TRC in terms of $u$ and $m$ which we change to rewriting $u = n - m$ such that the expression is only in terms of the total number of nodes and the number of marked nodes.

## C.1 TRC calculus

In the following we proof some preliminary lemmas for TRC that we will use in the later steps.

**Lemma 1 (Scalar multiplication)** *Let $A \subseteq \mathbb{R}^n$, a scalar $c \in \mathbb{R}$ and a vector $\boldsymbol{a}_0 \in \mathbb{R}^n$ then*

$$\mathfrak{R}_{m,n} \left( \{c\boldsymbol{a} + \boldsymbol{a}_0 : \boldsymbol{a} \in A\} \right) \leq |c| \mathfrak{R}_{m,n}(A)$$

*Proof.* Directly by construction. $\square$

**Lemma 2 (Addition)** *Let $A \subseteq \mathbb{R}^n, B \subseteq \mathbb{R}^n$ then*

$$\mathfrak{R}_{m,n}(A + B) = \mathfrak{R}_{m,n}(A) + \mathfrak{R}_{m,n}(B)$$

*Proof.* By construction and linearity of expectation. $\square$

**Lemma 3 (Convex hull)** *Let $A \subseteq \mathbb{R}^n$*
*and $A' = \left\{ \sum_{j=1}^{N} \alpha_j \boldsymbol{a}^{(j)} \ \middle| \ N \in \mathbb{N}, \ \forall j, \ \boldsymbol{a}^{(j)} \in A, \alpha_j \geq 0, \|\alpha\|_1 = 1 \right\}$ then*

$$\mathfrak{R}_{m,n}(A) = \mathfrak{R}_{m,n}(A').$$

*Proof.* The proof follows similar to the one for inductive Rademacher complexity (e.g. Shalev-Shwartz et al. (2014)). We first note that for any vector $\boldsymbol{v}$ the following holds:

$$\sup_{\alpha \geq 0 : \|\alpha\|_1 = 1} \sum_{j=1}^{N} \alpha_j \boldsymbol{v}_j = \max_{j} \boldsymbol{v}_j$$

Then:

$$
\begin{aligned}
\mathfrak{R}_{m,n}(A') &= Q \mathbb{E}_{\boldsymbol{\sigma}} \left[ \sup_{\alpha \geq 0 : \|\alpha\|_1 = 1} \sup_{\{\boldsymbol{a}^{(i)}\}_{i=1}^{N}} \sum_{i=1}^{n} \boldsymbol{\sigma}_i \sum_{j=1}^{N} \alpha_j \boldsymbol{a}_i^{(j)} \right] \\
&= Q \mathbb{E}_{\boldsymbol{\sigma}} \left[ \sup_{\alpha \geq 0 : \|\alpha\|_1 = 1} \sum_{j=1}^{N} \alpha_j \sup_{\boldsymbol{a}^{(j)}} \sum_{i=1}^{n} \boldsymbol{\sigma}_i \boldsymbol{a}_i^{(j)} \right] \\
&= Q \mathbb{E}_{\boldsymbol{\sigma}} \left[ \sup_{\boldsymbol{a} \in A} \sum_{i=1}^{n} \boldsymbol{\sigma}_i \boldsymbol{a}_i \right] \\
&= \mathfrak{R}_{m,n}(A)
\end{aligned}
$$

which concludes the proof. $\square$

**Lemma 4 (Contraction El-Yaniv et al. (2009))** *Let $A \subseteq \mathbb{R}^n$ be a set of vectors. Let $f(\,\cdot\,)$ and $g(\,\cdot\,)$ be real-value functions. Let $\boldsymbol{\sigma} = \{\sigma_i\}_{i=1}^{n}$ be Rademacher variables as defined in Definition 1. If for all $1 \leq i \leq n$ and any $\boldsymbol{a}, \boldsymbol{a}' \in A$, $|f(\boldsymbol{a}_i) - f(\boldsymbol{a}_i')| \leq |g(\boldsymbol{a}_i) - g(\boldsymbol{a}_i')|$ then*

$$\mathbb{E}_{\boldsymbol{\sigma}} \left[ \sum_{i=1}^{n} \sigma_i f(\boldsymbol{a}_i) \right] \leq \mathbb{E}_{\boldsymbol{\sigma}} \left[ \sum_{i=1}^{n} \sigma_i g(\boldsymbol{a}_i) \right]$$

*Extending this to Lipschitz continues functions. Let $v(\,\cdot\,)$ be a $L_v$-Lipschitz continues function such that $|v(f(\boldsymbol{a}_i)) - v(f(\boldsymbol{a}_i'))| \leq \frac{1}{L_v} |f(\boldsymbol{a}_i) - f(\boldsymbol{a}_i')|$. Now let the corresponding hypothesis classes be $\mathcal{F} \triangleq \{f(\cdot)\}, \mathcal{V} \triangleq \{v(f(\cdot))\}$ then*

$$\mathfrak{R}_{m,n}(\mathcal{V}) \leq \frac{1}{L_v} \mathfrak{R}_{m,n}(\mathcal{H}) \tag{18}$$

**Lemma 5 (Cardinality of finite sets)** *Let $A = \{\boldsymbol{a}_1, \cdots, \boldsymbol{a}_n\}$ be a finite set of vectors in $\mathbb{R}^d$ and let $\overline{\boldsymbol{a}} = \frac{1}{n} \sum_{i=1}^{n} \boldsymbol{a}_i$ then*

$$\mathfrak{R}_{m,n}(A) \leq \max_{\boldsymbol{a} \in A} \|\boldsymbol{a} - \overline{\boldsymbol{a}}\|_2 \sqrt{\frac{2 \log(n)}{d}}$$

*Proof.* The proof follows the general idea of the proof for *Massarts Lemma* (see e.g. Shalev-Shwartz et al. (2014)).

From Lemma 3 wlog. let $\overline{a} = 0$. Let $\lambda > 0$ and $A' = \{\lambda a_1, \cdots, \lambda a_n\}$. Therefore

$$\frac{1}{Q}\Re_{m,n}(A') = \mathbb{E}_{\sigma}\left[\max_{a \in A'}\langle\sigma, a\rangle\right]$$

$$= \mathbb{E}_{\sigma}\left[\log\left(\max_{a \in A'}\exp\left(\langle\sigma, a\rangle\right)\right)\right]$$

$$\leq \mathbb{E}_{\sigma}\left[\log\left(\sum_{a \in A'}\exp\left(\langle\sigma, a\rangle\right)\right)\right] \qquad \text{Jensen inequality}$$

$$\leq \log\left(\mathbb{E}_{\sigma}\left[\sum_{a \in A'}\exp\left(\langle\sigma, a\rangle\right)\right]\right) \qquad \sigma_i \text{ is i.i.d.}$$

$$= \log\left(\sum_{a \in A'}\prod_{i=1}\mathbb{E}_{\sigma_i}\left[\exp(\sigma_i a_i)\right]\right)$$

Bound $\mathbb{E}_{\sigma_i}\left[\exp(\sigma_i a_i)\right]$:

$$\mathbb{E}_{\sigma_i}\left[\exp(\sigma_i a_i)\right] = p\exp(1 a_i) + (1 - 2p)\exp(0 a_i) + p\exp(-1 a_i) \qquad \text{by definition of } \sigma_i$$

$$= (1 - 2p) + p\sum_{i=0}^{\infty}\frac{(-a)^i + a^i}{i!}$$

$$\leq \frac{1}{2}\sum_{i=0}^{\infty}\frac{(-a)^i + a^i}{i!} \qquad \text{as } p \leq \frac{1}{2}. \text{ Equality for } p = \frac{1}{2}.$$

$$= \frac{\exp(a_i) + \exp(-a_i)}{2}$$

$$\leq \exp\left(\frac{a_i^2}{2}\right)$$

Because

$$\frac{\exp(a) + \exp(-a)}{2} = \sum_{n=0}^{\infty}\frac{a^{2n}}{(2n)!} \leq \sum^{\infty}\frac{a^{2n}}{2^n n!} = \frac{a^{2n}}{2^n n!}\exp\left(\frac{a^2}{2}\right)$$

and $(2n)! \geq 2^n n! \ \forall n \geq 0$. Going back we now get:

$$\frac{1}{Q}\Re_{m,n}(A') \leq \log\left(\sum_{a \in A'}\prod_{i=1}\mathbb{E}_{\sigma_i}\left[\exp(\sigma_i a_i)\right]\right)$$

$$\leq \log\left(\sum_{a \in A'}\prod_{i=1}\exp\left(\frac{a_i^2}{2}\right)\right)$$

$$= \log\left(\sum_{a \in A'}\exp\left(\frac{\|a\|^2}{2}\right)\right)$$

$$\leq \log\left(|A'|\max_{a \in A'}\exp\left(\frac{\|a\|^2}{2}\right)\right)$$

$$= \log\left(|A'|\right) + \max_{a \in A'}\left(\frac{\|a\|^2}{2}\right)$$

By construction $\Re_{m,n}(A) = \frac{1}{\lambda}\Re_{m,n}(A')$ and therefore $\Re_{m,n} \leq \frac{1}{\lambda d}\left(\log(|A|) + \lambda^2\max_{a \in A'}\left(\frac{\|a\|^2}{2}\right)\right)$. By setting $\lambda = \sqrt{\frac{2\log(|A|)}{\max_{a \in A'}\|a\|^2}}$ and rearranging:

$$\Re_{m,n}(A) \leq \max_{a \in A}\|a - \overline{a}\|_2\sqrt{\frac{2\log(n)}{d}}$$

which concludes the proof. □

## C.2 Recursive bound on the TRC of single neurons

We start from the general GNN setup as defined as follows: Consider a class of GNNs defined over $K$ layers, with dimension of layer $k \in [K]$ being $d_k$ and $\boldsymbol{S} \in \mathbb{R}^{n \times n}$ the diffusion operator. Let $\phi, \psi$ be $L_\phi, L_\psi$-Lipschitz pointwise functions. Define:

$$g_{k+1} \triangleq \phi\left(\boldsymbol{b}_k + \boldsymbol{S} g_k\left(\boldsymbol{H}\right) \boldsymbol{W}_k\right),$$
$$g_0 \triangleq \boldsymbol{X}$$

and the hypothesis class over all such functions as

$$\mathcal{H}_{\mathcal{G}}^{\phi,\psi} \triangleq \left\{ h_{\mathcal{G}}^{\phi,\psi}(\boldsymbol{X}) = \psi\left(g_K \circ \cdots \circ g_0\right) \right\}.$$

From there we derive a recursive TRC bound depending on the previous layer.

Consider $g_{k+1} \triangleq \phi\left(\boldsymbol{b}_k + \boldsymbol{S} g_k\left(\boldsymbol{H}\right) \boldsymbol{W}_k\right)$, $k \in \{1, \cdots, K\}$. Now we define the function class over one neuron as

$$\mathcal{H}_{\mathcal{G}}^{\phi} \triangleq \left\{ h_{\mathcal{G}}^{\phi}(\boldsymbol{H}) = \phi\left( \boldsymbol{b}_i + \sum_l^{d_k} \boldsymbol{W}_{lj} \sum_t^{n} \boldsymbol{S}_{it} g(\boldsymbol{H})_{lj} \right) \; \middle| \; g \in \mathcal{F}, \|\boldsymbol{b}\|_1 \leq \beta \right\}$$

where $\mathcal{F}$ is the class of $\mathbb{R}^{n \times d_k} \to \mathbb{R}$, including the zero function. Then with $\boldsymbol{W}_{\cdot j} \triangleq [\boldsymbol{W}_{1j}, \cdots, \boldsymbol{W}_{d_k j}]^\top$:

$$\mathfrak{R}_{m,n}(\mathcal{H}_{\mathcal{G}}^{\phi}) \leq 2 L_\phi \left( \beta Q(n) + \|\boldsymbol{S}\|_\infty \|\boldsymbol{W}_{\cdot j}\|_1 \mathfrak{R}_{m,n}(\mathcal{F}) \right)$$

*Proof.*

By Lemma 4 and Lemma 2 we get

$$\mathfrak{R}_{m,n}(\mathcal{H}_{\mathcal{G}}^{\phi}) \leq L_\phi \left( \mathfrak{R}_{m,n}(\mathcal{H}_{lin}) + \mathfrak{R}_{m,n}(\mathcal{H}_{bias}) \right)$$

where

$$\mathcal{H}_{lin} \triangleq \left\{ h_{lin}(\boldsymbol{H}) = \sum_l^{d_k} \boldsymbol{W}_{lj} \sum_t^{n} \boldsymbol{S}_{it} g(\boldsymbol{H})_{lj} \; \middle| \; g \in \mathcal{F}, \|\boldsymbol{W}_{\cdot j}\|_1 \leq \omega \right\}$$
$$\mathcal{H}_{bias} \triangleq \left\{ h_{bias}(\boldsymbol{H}) = \boldsymbol{b} \; \middle| \; |b| \leq \beta \right\}$$

with $\boldsymbol{W}_{\cdot j} \triangleq [\boldsymbol{W}_{1j}, \cdots, \boldsymbol{W}_{d_k j}]^\top$. Bounding terms individually.

Bound $\mathfrak{R}_{m,n}(\mathcal{H}_{lin})$

We start by rewriting the linear term. For readability $g_{lj} := g(\boldsymbol{H})_{lj}$

$$\boldsymbol{H}_{ij} = \sum_l^{d_k} \boldsymbol{W}_{lj} \sum_t^{n} \boldsymbol{S}_{it} g_{lj}$$
$$= \underbrace{\boldsymbol{W}_{1j} \boldsymbol{S}_{i1} g_{1j} + \cdots + \boldsymbol{W}_{1j} \boldsymbol{S}_{in} g_{1j}}_{\boldsymbol{W}_{1j} g_{1j} \left( \sum_t^n \boldsymbol{S}_{it} \right)} + \underbrace{\boldsymbol{W}_{2j} \boldsymbol{S}_{i1} g_{2j} + \cdots + \boldsymbol{W}_{2j} \boldsymbol{S}_{in} g_{2j}}_{\boldsymbol{W}_{2j} g_{2j} \left( \sum_t^n \boldsymbol{S}_{it} \right)} + \cdots$$
$$\text{with } \sum_t^{n} \boldsymbol{S}_{it} \leq \|\boldsymbol{S}\|_\infty$$
$$\leq \|\boldsymbol{S}\|_\infty \left( \sum_l^{d_k} \boldsymbol{W}_{1j} g_{1j} \right)$$

Now we define

$$\widetilde{\mathcal{H}}_{lin} \triangleq \left\{ h_{lin}(\boldsymbol{H}) = \sum_{l}^{d_k} \boldsymbol{W}_{lj} g(\boldsymbol{H})_{lj} \,\middle|\, g \in \mathcal{F}, \|\boldsymbol{W}_{\cdot j}\|_1 \leq \omega \right\}$$

$$\widetilde{\mathcal{H}}'_{lin} \triangleq \left\{ h_{lin}(\boldsymbol{H}) = \sum_{l}^{d_k} \boldsymbol{W}_{lj} g(\boldsymbol{H})_{lj} \,\middle|\, g \in \mathcal{F}, \|\boldsymbol{W}_{\cdot j}\|_1 = \omega \right\}$$

and since $\|\boldsymbol{S}\|_\infty$ is constant we get by Lemma 1

$$\mathfrak{R}_{m,n}(\mathcal{H}_{lin}) \leq \|\boldsymbol{S}\|_\infty \mathfrak{R}_{m,n}(\widetilde{\mathcal{H}}_{lin}).$$

To further bound $\mathfrak{R}_{m,n}(\widetilde{\mathcal{H}}_{lin})$ we can a similar process then for standard deep neural networks with slight deviation on the indexing of the weight matrix.

Let $\mathrm{Hull}\,(\cdot)$ be a convex hull. In the first step we show that

$$\mathfrak{R}_{m,n}(\widetilde{\mathcal{H}}_{lin}) = \omega \mathfrak{R}_{m,n}(\mathrm{Hull}\,(\mathcal{F} - \mathcal{F}))$$

where $\mathcal{F} - \mathcal{F} \triangleq \{f - f', \; f \in \mathcal{F}, f' \in \mathcal{F}\}$. Note that the maximum over all function over $\boldsymbol{W}_{il}$ with constraint $\|\boldsymbol{W}_{\cdot j}\|_1 \leq \omega$ is achieved for $\|\boldsymbol{W}_{\cdot j}\|_1 = \omega$ then

$$\mathfrak{R}_{m,n}(\widetilde{\mathcal{H}}_{lin}) = \mathfrak{R}_{m,n}(\widetilde{\mathcal{H}}'_{lin})$$

Let $\boldsymbol{0}$ be the zero function. Then for $\|\boldsymbol{W}_{\cdot j}\|_1 = 1$:

$$\sum_{l} \boldsymbol{W}_{lj} g_{lj} = \sum_{l:\boldsymbol{W}_{lj} \geq 0} \boldsymbol{W}_{lj}(g_{lj} - \boldsymbol{0}) + \sum_{l:\boldsymbol{W}_{lj} < 0} |\boldsymbol{W}_{lj}|(\boldsymbol{0} - g_{lj})$$

which is $\mathrm{Hull}\,(\mathcal{F} - \mathcal{F})$. Combining the above results we get:

$$
\begin{aligned}
\mathfrak{R}_{m,n}(\mathcal{H}_{lin}) &\leq \|\boldsymbol{S}\|_\infty \omega \mathfrak{R}_{m,n}\left(\widetilde{\mathcal{H}}_{lin}\right) \\
&= \|\boldsymbol{S}\|_\infty \omega \mathfrak{R}_{m,n}\left(\mathrm{Hull}\,(\mathcal{F} - \mathcal{F})\right) \\
&= \|\boldsymbol{S}\|_\infty \omega \mathfrak{R}_{m,n}\left(\mathcal{F} - \mathcal{F}\right) \\
&= \|\boldsymbol{S}\|_\infty \omega \left(\mathfrak{R}_{m,n}\left(\mathcal{F}\right) + \mathfrak{R}_{m,n}\left(-\mathcal{F}\right)\right) \qquad \text{Lemma 2} \\
&= 2 \|\boldsymbol{S}\|_\infty \omega \mathfrak{R}_{m,n}\left(\mathcal{F}\right) \qquad\qquad\qquad\quad \text{Lemma 1}
\end{aligned}
$$

which concludes this part of the proof.

Bound $\mathfrak{R}_{m,n}(\mathcal{H}_{bias})$

Start by writing out $\mathfrak{R}_{m,n}(\cdot)$

$$\mathfrak{R}_{m,n}(\mathcal{H}_{bias}) = Q\mathbb{E}_{\boldsymbol{\sigma}}\left[\sup_{\boldsymbol{b}:|\boldsymbol{b}|\leq\beta} \boldsymbol{b}\sum_{i=1}^{n}\boldsymbol{\sigma}_i\right]$$

$$\leq Q\mathbb{E}_{\boldsymbol{\sigma}}\left[\sup_{\boldsymbol{b}:|\boldsymbol{b}|\leq\beta} |\boldsymbol{b}|\left|\sum_{i=1}^{n}\boldsymbol{\sigma}_i|\right|\right]$$

$$= \beta Q\mathbb{E}_{\boldsymbol{\sigma}}\left[\left|\sum_{i=1}^{n}\boldsymbol{\sigma}_i\right|\right]$$

$$\leq \beta Q\mathbb{E}_{\boldsymbol{\sigma}}\left[\sum_{i=1}^{n}|\boldsymbol{\sigma}_i|\right]$$

$$\leq \beta Q\sum_{i=1}^{n}\mathbb{E}_{\boldsymbol{\sigma}}\left[|\boldsymbol{\sigma}_i|\right]$$

$$\leq \beta Q\sum_{i=1}^{n}2p$$

$$\leq \beta Q(n)2p$$

$$\leq \beta Q(n)2$$

which concludes this part of the proof. Combining the two bounds gives:

$$\mathfrak{R}_{m,n}(\mathcal{H}_{\mathcal{G}}^{\phi}) \leq 2L_{\phi}\left(\beta Q(n) + \|\boldsymbol{S}\|_{\infty}\|\boldsymbol{W}_{\cdot j}\|_1 \mathfrak{R}_{m,n}(\mathcal{F})\right)$$

concluding the proof of Proposition 2. $\qquad\square$

### C.3 Bound on the TRC for the first layer

Define the hypothesis class over the function of the first layer $g_0$ as:

$$\mathcal{H}_{\mathcal{G}}^{\phi} \triangleq \left\{ h_{\mathcal{G}}^{\phi}(\boldsymbol{X}) = \phi\left(\boldsymbol{b} + \boldsymbol{S}\boldsymbol{X}\boldsymbol{W}_1\right)\right\}$$

then the TRC is give by

$$\mathfrak{R}_{m,n}(\mathcal{H}_{\mathcal{G}}^{\phi}) \leq L_{\phi}\left(\|\boldsymbol{b}\|_1 Q(n)2 + Q\|\boldsymbol{W}_1\|_{\infty}\|\boldsymbol{S}\boldsymbol{X}\|_{2\to\infty}\sqrt{\frac{2\log(n)}{d}}\right)$$

*Proof.* Frist similar to Proposition 2 we use Lemma 2 and Lemma 4

$$\mathfrak{R}_{m,n}(\mathcal{H}_{\mathcal{G}}^{\phi}) \leq L_{\phi}\left(\mathfrak{R}_{m,n}(\mathcal{H}_{lin}) + \mathfrak{R}_{m,n}(\mathcal{H}_{bias})\right)$$

As before $\mathfrak{R}_{m,n}(\mathcal{H}_{bias}) \leq \beta Q(n)2p$. In this case we define the linear term as

$$\mathcal{H}_{lin} \triangleq \{h_{lin}(\boldsymbol{X}) = \boldsymbol{S}\boldsymbol{X}\boldsymbol{W}\}.$$

Bounding the TRC of $\mathcal{H}_{lin}$

$$\mathfrak{R}_{m,n}(\mathcal{H}_{lin}) = Q\mathbb{E}_{\boldsymbol{\sigma}}\left[\sup_{\boldsymbol{W}:\|\boldsymbol{W}\|_{\infty}\leq\omega} \boldsymbol{\sigma}^{\top}\boldsymbol{S}\boldsymbol{X}\boldsymbol{W}\right]$$

$$\leq Q\|\boldsymbol{W}\|_{\infty}\mathbb{E}_{\boldsymbol{\sigma}}\left[\|\boldsymbol{\sigma}^{\top}\boldsymbol{S}\boldsymbol{X}\|_{\infty}\right]$$

To bound $\mathbb{E}_{\boldsymbol{\sigma}}\left[\left\|\boldsymbol{\sigma}^{\top}\boldsymbol{S}\boldsymbol{X}\right\|_{\infty}\right]$ we define $\boldsymbol{t}_i = (x_{1j},\ldots,x_{nj})^{\top}$ and $T = \{t_1,\ldots,t_n\}, T_- = \{-t_1,\ldots,-t_n\}$. Therefore

$$
\begin{aligned}
\mathbb{E}_{\boldsymbol{\sigma}}\left[\left\|\boldsymbol{\sigma}^{\top}\boldsymbol{S}\boldsymbol{X}\right\|_{\infty}\right] &\leq \mathbb{E}_{\boldsymbol{\sigma}}\left[\max_{t\in T}|\boldsymbol{\sigma}^{\top}\boldsymbol{S}\boldsymbol{t}|\right] \\
&= \mathbb{E}_{\boldsymbol{\sigma}}\left[\max_{t\in T\cup T_-}\boldsymbol{\sigma}^{\top}\boldsymbol{S}\boldsymbol{t}\right] \\
&\leq \max_{t\in T\cup T_-}\|\boldsymbol{S}\boldsymbol{t}\|_2\sqrt{\frac{2\log(n)}{d}} \qquad \text{Lemma 5} \\
&= \|\boldsymbol{S}\boldsymbol{t}\|_{2\to\infty}\sqrt{\frac{2\log(n)}{d}}
\end{aligned}
$$

Combining with the above results gives

$$
\mathfrak{R}_{m,n}(\mathcal{H}_{lin}) \leq Q\,\|\boldsymbol{W}\|_{\infty}\,\|\boldsymbol{S}\boldsymbol{t}\|_{2\to\infty}\sqrt{\frac{2\log(n)}{d}}.
$$

Taking the bound on the bias term into considerations gives the final bound and concludes the proof of Proposition 3. $\qquad\square$

### C.4 Additional notes on the remarks related to Theorem 1

**Influence of the graph information: Empty and fully-connected graph.** To be able to analyse the influence of the graph information we can note that the graph information comes into play through $\|\boldsymbol{S}\boldsymbol{X}\|_{2\to\infty}$. We can rewrite this expression as $\|\boldsymbol{S}\boldsymbol{X}\|_{2\to\infty} = \max_j \sqrt{\sum_i (\boldsymbol{S}\boldsymbol{X})_{ij}^2}$ and then by replacing $\boldsymbol{S}$ with the empty ($\boldsymbol{A} = K_{\mathcal{G}}$) and the complete graph ($\boldsymbol{A} = \mathbb{I}$) gives: $\|K_{\mathcal{G}}\boldsymbol{X}\|_{2\to\infty} = \max_j \frac{1}{\sqrt{n}}\sqrt{\left(\sum_k \boldsymbol{X}_{kj}\right)^2}$, and $\|\mathbb{I}\boldsymbol{X}\|_{2\to\infty} = \max_j \sqrt{\sum_k \boldsymbol{X}_{kj}^2}$ and since $\left(\sum_k \boldsymbol{X}_{kj}\right)^2 \leq n\,\|\boldsymbol{X}_{\cdot j}\|_2^2$ it follows that $\mathfrak{R}(\mathcal{H}_{K_{\mathcal{G}}}^{\phi}) \leq \mathfrak{R}(\mathcal{H}_{\mathbb{I}}^{\phi})$ which is consistent with the observation obtained from the VC-Dimension bound. In both cases the complexity measure of the fully connected graph is lower then the if we would not consider graph information.

**Influence of the graph information: $b$-regular graph.** Now consider a setup that incorporates a larger number of graphs. Assume $\boldsymbol{S} := \boldsymbol{D}^{-\frac{1}{2}}(\boldsymbol{A} + \mathbb{I})\boldsymbol{D}^{-\frac{1}{2}}$ and that we only consider the graph information (e.g. $\boldsymbol{X} = \mathbb{I}$), then for a $b$-regular graph (a graph where every vertex has degree $b$) we can write $\|\boldsymbol{S}\mathbb{I}\|_{2\to\infty} = \max_j \|\boldsymbol{S}_{\cdot j}\|_2 = \sqrt{\sum_{i\sim j}\frac{1}{\boldsymbol{D}_i\boldsymbol{D}_j}} = \frac{1}{\sqrt{b}} < 1$. Therefore adding graph information results in $\mathfrak{R}(\mathcal{H}_{\mathcal{G}}^{\phi}) \leq \mathfrak{R}(\mathcal{H}_{\mathbb{I}}^{\phi})$ and therefore the complexity resulting in not using graph information upper bounds the complexity that results if we consider graph information.

# D    Proof Theorem 3 — TRC for Residual GNNs

Recall the setup for residual connections as defined in the main paper where we can now write the layer wise propagation rule as

$$g_{k+1} \triangleq \phi\left((1-\alpha)\left(\boldsymbol{b}_k + \boldsymbol{S}g_k\left(\boldsymbol{H}\right)\boldsymbol{W}_k\right) + \alpha g_0\left(\boldsymbol{H}\right)\right), \qquad \text{with } \alpha \in (0,1).$$

We can now derive a generalization error bound similar to the one given in Theorem 1 for the Residual network. As most of the steps are the same we will only remark the main changes. Recall that for the vanilla case we considered

$$\mathfrak{R}_{m,n}(\mathcal{H}_{\mathcal{G}}^{\phi}) \leq L_{\phi}\left(\mathfrak{R}_{m,n}(\mathcal{H}_{lin}) + \mathfrak{R}_{m,n}(\mathcal{H}_{bias})\right)$$

and by Lemma 2 and Lemma 1 we obtain a similar bound for the Residual network as

$$\mathfrak{R}_{m,n}(\mathcal{H}_{\mathcal{G}}^{\phi}) \leq L_{\phi}\left((1-\alpha)\mathfrak{R}_{m,n}(\mathcal{H}_{lin}) + (1-\alpha)\mathfrak{R}_{m,n}(\mathcal{H}_{bias}) + \alpha\mathfrak{R}_{m,n}(\mathcal{H}_{\boldsymbol{X}})\right).$$

The bounds for $\mathfrak{R}_{m,n}(\mathcal{H}_{lin})$ and $\mathfrak{R}_{m,n}(\mathcal{H}_{bias})$ are as derived in section C. $\mathfrak{R}_{m,n}(\mathcal{H}_{\boldsymbol{X}})$ can be bound as

$$\mathfrak{R}_{m,n}(\mathcal{H}_{\boldsymbol{X}}) \leq 2Q\left\|\boldsymbol{X}\right\|_{\infty} n$$

Where the proof follows analogous to the one for the *bias term*, $\mathfrak{R}_{m,n}(\mathcal{H}_{bias}$.

Again with recursively applying the bounds for each layer and combining it with the bound on the first layer results in the full TRC bound. Consider a Residual network as defined in (13) and $\mathcal{H}_{\mathcal{G}}^{\phi,\beta,\omega} \subset \mathcal{H}_{\mathcal{G}}^{\phi}$ such that the trainable parameters satisfy $\left\|\boldsymbol{b}_k\right\|_1 \leq \beta$ and $\left\|\boldsymbol{W}_k\right\|_{\infty} \leq \omega$ for every $k \in [K]$. Then with $\alpha \in (0,1)$ and $c_1 \triangleq 2L_{\phi}\beta$, $c_2 \triangleq 2L_{\phi}\omega$, $c_3 \triangleq L_{\phi}\omega\sqrt{2/d}$ the TRC of the restricted class or Residual GNNs is bounded as

$$\begin{aligned}
\mathfrak{R}_{m,n}(\mathcal{H}_{\mathcal{G}}^{\phi,\beta,\omega}) \leq{}& \frac{((1-\alpha)c_1 + \alpha 2L_{\phi}\left\|\boldsymbol{X}\right\|_{\infty})n^2}{m(n-m)}\left(\sum_{k=0}^{K-1}(1-\alpha)c_2^k\left\|\boldsymbol{S}\right\|_{\infty}^k\right) \\
&+ \alpha 2L_{\phi}\left\|\boldsymbol{X}\right\|_{\infty} + (1-\alpha)c_3 c_2^K\left\|\boldsymbol{S}\right\|_{\infty}^K\left\|\boldsymbol{S}\boldsymbol{X}\right\|_{2\to\infty}\sqrt{\log(n)}
\end{aligned}$$

# E  Proof Theorem 2 — Expected TRC for GNNs under SBM

## E.1  Setup (recap from the main paper)

We assume that the node features are sampled latent true classes, given a $\boldsymbol{z} = (z_1, \ldots, z_n) \in \{\pm 1\}^n$. The node features are sampled from a Gaussian mixture model (GMM), that is, feature for node-$i$ is sampled as $\boldsymbol{x}_i \sim \mathcal{N}(z_i \boldsymbol{\mu}, \sigma^2 \mathbb{I})$ for some $\boldsymbol{\mu} \in \mathbb{R}^d$ and $\sigma \in (0, \infty)$. We express this in terms of $\boldsymbol{X}$ as

$$\boldsymbol{X} = \mathcal{X} + \boldsymbol{\epsilon} \in \mathbb{R}^{n \times d}, \qquad \text{where } \mathcal{X} = \boldsymbol{z} \boldsymbol{\mu}^\top \text{ and } \boldsymbol{\epsilon} = (\epsilon_{ij})_{i \in [n], j \in [d]} \stackrel{i.i.d.}{\sim} \mathcal{N}(0, \sigma^2).$$

We refer to above as $\boldsymbol{X} \sim 2\text{GMM}$. On the other hand, we assume that graph has two latent communities, characterised by $\boldsymbol{y} \in \{\pm 1\}^n$. The graph is generated from a stochastic block model with two classes (2SBM), where edges $(i, j)$ are added independently with probability $p \in (0, 1]$ if $y_i = y_j$, and with probability $q < [0, p)$ if $y_i \neq y_j$. In other words, we define the random adjacency $\boldsymbol{A} \sim 2\text{SBM}$ as a symmetric binary matrix with $\boldsymbol{A}_{ii} = 0$, and $(\boldsymbol{A}_{ij})_{i<j}$ indenpendent such that

$$\boldsymbol{A}_{ij} \sim \text{Bernoulli}(\mathcal{A}_{ij}), \qquad \text{where } \mathcal{A} = \frac{p+q}{2} \mathbf{1} \mathbf{1}^\top + \frac{p-q}{2} \boldsymbol{y} \boldsymbol{y}^\top - p \mathbb{I}.$$

The choice of two different latent classes $\boldsymbol{z}, \boldsymbol{y} \in \{\pm 1\}^n$ allows study of the case where the graph and feature information of do not align completely. We use $\Gamma = |\boldsymbol{y}^\top \boldsymbol{z}| \in [0, n]$ to quantify this alignment. Assuming $\boldsymbol{y}, \boldsymbol{z}$ are both balanced, that is, $\sum_i y_i = \sum_i z_i = 0$.

In addition the TRC is given by Theorem 1:

Consider $\mathcal{H}_{\mathcal{G}}^{\phi, \beta, \omega} \subseteq \mathcal{H}_{\mathcal{G}}^\phi$ such that the trainable parameters satisfy $\|\boldsymbol{b}_k\|_1 \leq \beta$ and $\|\boldsymbol{W}_k\|_\infty \leq \omega$ for every $k \in [K]$. The transductive Randemacher complexity (TRC) of the restricted hypothesis class is bounded as

$$\Re_{m,n}(\mathcal{H}_{\mathcal{G}}^{\phi, \beta, \omega}) \leq \frac{c_1 n^2}{m(n-m)} \left( \sum_{k=0}^{K-1} c_2^k \|\boldsymbol{S}\|_\infty^k \right) + c_3 c_2^K \|\boldsymbol{S}\|_\infty^K \|\boldsymbol{S} \boldsymbol{X}\|_{2 \to \infty} \sqrt{\log(n)},$$

where $c_1 \triangleq 2 L_\phi \beta$, $c_2 \triangleq 2 L_\phi \omega$, $c_3 \triangleq L_\phi \omega \sqrt{2/d}$ and $L_\phi$ is Lipschitz constant for activation $\phi$.

## E.2  Main Proof

From the above bound we can note that to derive the TRC in expectation we have to compute $\mathbb{E}\left[ \|\boldsymbol{S}\|_\infty^k \right]$ and $\mathbb{E}\left[ \|\boldsymbol{S}\|_\infty^k \|\boldsymbol{S} \boldsymbol{X}\|_{2 \to \infty} \right]$ where we can decompose the latter as follows

$$\begin{aligned}
\mathbb{E}\left[ \|\boldsymbol{S}\|_\infty^k \|\boldsymbol{S} \boldsymbol{X}\|_{2 \to \infty} \right] \leq & \mathbb{E}\left[ \|\boldsymbol{S}\|_\infty^k \|\mathcal{S} \mathcal{X}\|_{2 \to \infty} \right] \\
& + \mathbb{E}\left[ \|\boldsymbol{S}\|_\infty^k \|(\boldsymbol{S} - \mathcal{S}) \mathcal{X}\|_{2 \to \infty} \right] + \mathbb{E}\left[ \|\boldsymbol{S}\|_\infty^k \|\boldsymbol{S}(\boldsymbol{X} - \mathcal{X})\|_{2 \to \infty} \right] \\
\leq & \|\mathcal{S} \mathcal{X}\|_{2 \to \infty} \mathbb{E}\left[ \|\boldsymbol{S}\|_\infty^k \right] \\
& + \sqrt{\mathbb{E}\left[ \|\boldsymbol{S}\|_\infty^{2k} \right]} \sqrt{\mathbb{E}\left[ \|(\boldsymbol{S} - \mathcal{S}) \mathcal{X}\|_{2 \to \infty}^2 \right]} \\
& + \sqrt{\mathbb{E}\left[ \|\boldsymbol{S}\|_\infty^{2k} \right]} \sqrt{\mathbb{E}\left[ \|(\boldsymbol{X} - \mathcal{X}) \boldsymbol{S}\|_{2 \to \infty}^2 \right]}
\end{aligned}$$

where the second inequality follows from noting that $\|\mathcal{S} \mathcal{X}\|_{2 \to \infty}$ is deterministic and does not depend on the expectation and the decomposition of the last two terms follows from using Cauchy-Schwarz inequality.

Table 1 gives an overview over the bounds on the different terms, where the individual entries are derived in section E.3.

Table 1: Overview over different concentration bounds for *self loop* and *degree normalization*. Let $C = (1 + o(1))$

| | Self Loop | Degree Normalized |
|---|---|---|
| E.3.4: $\|\mathcal{S}\mathcal{X}\|_{2\to\infty}$ | $C\|\boldsymbol{\mu}\|_\infty n\left(1 + \left(\frac{p-q}{2}\right)^2\Gamma^2\right)$ | $C\|\boldsymbol{\mu}\|_\infty \frac{\left(1 + \left(\frac{p-q}{2}\right)^2\Gamma^2\right)}{\left(\frac{p+q}{2}\right)}$ |
| E.3.1: $\mathbb{E}\left[\|(\boldsymbol{S} - \mathcal{S})\,\mathcal{X}\|_{2\to\infty}^2\right]$ | $Cn^2 p\|\boldsymbol{\mu}\|_\infty$ | $C\frac{n\ln(n)}{1+(n-1)q}\|\boldsymbol{\mu}\|_\infty$ |
| E.3.2: $\mathbb{E}\left[\|(\boldsymbol{X} - \mathcal{X})\,\boldsymbol{S}\|_{2\to\infty}^2\right]$ | $Cn^2 p\sigma^2(1 + 2\ln d)$ | $C\frac{1}{q}$ |
| E.3.3: $\mathbb{E}\left[\|\boldsymbol{S}\|_\infty^k\right]$ | $(Cnp)^k$ | $\left(C\frac{p}{q}\right)^{\frac{k}{2}}$ |

## E.3   Concentration Bounds

### E.3.1   Bound $\mathbb{E}\left[\|(\boldsymbol{S} - \mathcal{S})\,\mathcal{X}\|_{2\to\infty}\right]$

We first note that:

$$
\begin{aligned}
\|(\boldsymbol{S} - \mathcal{S})\,\mathcal{X}\|_{2\to\infty} &= \left\|(\boldsymbol{S} - \mathcal{S})\,\boldsymbol{z}\boldsymbol{\mu}^\top\right\|_{2\to\infty} && \text{by definition of } \mathcal{X} \\
&= \max_j \left\|(\boldsymbol{S} - \mathcal{S})\,\boldsymbol{z}\boldsymbol{\mu}_j^\top\right\|_2 && \text{by definition of } \|\,\cdot\,\|_{2\to\infty} \\
&= \|(\boldsymbol{S} - \mathcal{S})\,\boldsymbol{z}\|_2\,\|\boldsymbol{\mu}\|_\infty && (19)
\end{aligned}
$$

and we only have to compute the expectation of $\|(\boldsymbol{S} - \mathcal{S})\,\boldsymbol{z}\|_2$ as $\|\boldsymbol{\mu}\|_\infty$ is deterministic. Taking the expectation:

$$
\begin{aligned}
\mathbb{E}\left[\|(\boldsymbol{S} - \mathcal{S})\,\boldsymbol{z}\|_2\right] &\leq \sqrt{\mathbb{E}\left[\boldsymbol{z}^\top (\boldsymbol{S} - \mathcal{S})^\top (\boldsymbol{S} - \mathcal{S})\,\boldsymbol{z}\right]} \\
&= \left(\sum_{ij} \boldsymbol{z}_i \boldsymbol{z}_j \sum_k \mathbb{E}\left[(\boldsymbol{S} - \mathcal{S})_{ki}\,(\boldsymbol{S} - \mathcal{S})_{kj}\right]\right)^{\frac{1}{2}} && (20)
\end{aligned}
$$

where (20) follows from the fact that $\boldsymbol{z}$ is deterministic. From this expression we can now consider the self loop and degree normalized case for the diffusion operator.

Case 1: Self loop.

$\sum_k \mathbb{E}\left[(\boldsymbol{S} - \mathcal{S})_{ki}\,(\boldsymbol{S} - \mathcal{S})_{kj}\right]$ in (20) now becomes $\sum_k \mathbb{E}\left[(\boldsymbol{A} - \mathcal{A})_{ki}\,(\boldsymbol{A} - \mathcal{A})_{kj}\right]$ where we distinguish two cases:

$$
\begin{aligned}
i \neq j &\quad\Rightarrow \boldsymbol{A}_{ki} \text{ and } \boldsymbol{A}_{kj} \text{ are independent} \Rightarrow \mathbb{E}\left[(\boldsymbol{A} - \mathcal{A})_{ki}\,(\boldsymbol{A} - \mathcal{A})_{kj}\right] = 0 \\
i = j &\quad\Rightarrow \mathbb{E}\left[(\boldsymbol{A} - \mathcal{A})_{ki}\,(\boldsymbol{A} - \mathcal{A})_{ki}\right] = \mathrm{Var}(\boldsymbol{A}_{ki}) = \mathcal{A}_{ki}(1 - \mathcal{A}_{ki})
\end{aligned}
$$

Therefore (20) becomes

$$\mathbb{E}\left[\|(\boldsymbol{S} - \mathcal{S})\,\boldsymbol{z}\|_2\right] \le \left(\sum_i z_i^2 \sum_k \mathcal{A}_{ki}(1 - \mathcal{A}_{ki})\right)^{\frac{1}{2}}$$

$$= \left(\sum_{ik} \mathcal{A}_{ki}(1 - \mathcal{A}_{ki})\right)^{\frac{1}{2}} \qquad \because z_i^2 = 1$$

$$\le \left(\sum_{ik} \mathcal{A}_{ki}\right)^{\frac{1}{2}}$$

$$\le \left(n^2 \frac{p+q}{2}\right)^{\frac{1}{2}}$$

$$= n\sqrt{\frac{p+q}{2}}$$

and giving us the final bound as using the above in (19):

$$\mathbb{E}\left[\|(\boldsymbol{S} - \mathcal{S})\,\mathcal{X}\|_{2\to\infty}\right] \le n\sqrt{\frac{p+q}{2}}\,\|\boldsymbol{\mu}\|_\infty$$

Case 2: Degree normalized.

Note that for this section we initially considered an extension of the degree normalized model where the self loop is weighted by $\gamma$. For the final version however we set $\gamma = 1$.

As before first note that:

$$\|(\boldsymbol{S} - \mathcal{S})\,\mathcal{X}\|_{2\to\infty} = \left\|(\boldsymbol{S} - \mathcal{S})\,\boldsymbol{z}\boldsymbol{\mu}^\top\right\|_{2\to\infty}$$

$$= \max_j \left\|(\boldsymbol{S} - \mathcal{S})\,\boldsymbol{z}\boldsymbol{\mu}_j^\top\right\|_2$$

$$= \|(\boldsymbol{S} - \mathcal{S})\,\boldsymbol{z}\|_2\,\|\boldsymbol{\mu}\|_\infty \qquad (21)$$

and we only have to compute the expectation of $\|(\boldsymbol{S} - \mathcal{S})\,\boldsymbol{z}\|_2$ in (21). To bound this term we start by defining:

$$\mathcal{S} \triangleq (\mathcal{D} + \gamma\mathbb{I})^{-\frac{1}{2}}(\mathcal{A} + \gamma\mathbb{I})(\mathcal{D} + \gamma\mathbb{I})^{-\frac{1}{2}}$$

$$\boldsymbol{S} \triangleq (\boldsymbol{D} + \gamma\mathbb{I})^{-\frac{1}{2}}(\boldsymbol{A} + \gamma\mathbb{I})(\boldsymbol{D} + \gamma\mathbb{I})^{-\frac{1}{2}}$$

$$\overline{\boldsymbol{S}} \triangleq (\mathcal{D} + \gamma\mathbb{I})^{-\frac{1}{2}}(\boldsymbol{A} + \gamma\mathbb{I})(\mathcal{D} + \gamma\mathbb{I})^{-\frac{1}{2}}$$

such that we can write:

$$\|(\boldsymbol{S} - \mathcal{S})\,\boldsymbol{z}\|_2 \le \left\|(\boldsymbol{S} - \overline{\boldsymbol{S}})\,\boldsymbol{z}\right\|_2 + \left\|(\overline{\boldsymbol{S}} - \mathcal{S})\,\boldsymbol{z}\right\|_2 \qquad (22)$$

and bound the two terms separately:

**Bound first term in** (22): $\left\|(\overline{\boldsymbol{S}} - \mathcal{S})\,\boldsymbol{z}\right\|_2$

First we note that:

$$\left\|(\overline{\boldsymbol{S}} - \mathcal{S})\,\boldsymbol{z}\right\|_2 \le \left\|(\mathcal{D} + \gamma\mathbb{I})^{-\frac{1}{2}}(\boldsymbol{A} - \mathcal{A})(\mathcal{D} + \gamma\mathbb{I})^{-\frac{1}{2}}\boldsymbol{z}\right\|_2$$

and therefore

$$\mathbb{E}\left[\left\|\left(\overline{\boldsymbol{S}} - \mathcal{S}\right)\boldsymbol{z}\right\|_2\right] \leq \left(\mathbb{E}\left[\boldsymbol{z}^\top(\mathcal{D}+\gamma\mathbb{I})^{-\frac{1}{2}}(\boldsymbol{A}-\mathcal{A})(\mathcal{D}+\gamma\mathbb{I})^{-1}(\boldsymbol{A}-\mathcal{A})(\mathcal{D}+\gamma\mathbb{I})^{-\frac{1}{2}}\boldsymbol{z}\right]\right)^{-\frac{1}{2}}$$

$$= \left(\sum_{i,j}\frac{\boldsymbol{z}_i\boldsymbol{z}_j}{\sqrt{(\gamma+\mathcal{D}_{ii})(\gamma+\mathcal{D}_{jj})}}\underbrace{\sum_{k\neq i,j}\frac{\mathbb{E}\left[(\boldsymbol{A}-\mathcal{A})_{ki}(\boldsymbol{A}-\mathcal{A})_{kj}\right]}{\gamma+\mathcal{D}_{kk}}}_{\text{term 2}}\right)^{-\frac{1}{2}} \quad (23)$$

$$\leq \left(\sum_i \frac{\boldsymbol{z}_i^2}{\gamma+\mathcal{D}_{ii}}\cdot\frac{\mathcal{D}_{ii}}{\gamma+(n-1)q}\right)^{-\frac{1}{2}} \quad (24)$$

$$\leq \left(\frac{n}{\gamma+(n-1)q}\right)^{-\frac{1}{2}} \qquad \because \boldsymbol{z}_i^2 = 1$$

Where the step form (23) to (24) follows by bounding (23), *term 2* as follows. For $i \neq j$ the expression is zero. Otherwise for $i = j$:

$$\sum_{k\neq i,j}\frac{\mathbb{E}\left[(\boldsymbol{A}-\mathcal{A})_{ki}(\boldsymbol{A}-\mathcal{A})_{kj}\right]}{\gamma+\mathcal{D}_{kk}} = \sum_{k\neq i}\frac{\text{Var}(\boldsymbol{A}_{ki})}{\gamma+\mathcal{D}_{kk}}$$

$$= \sum_{k\neq i}\frac{\mathcal{A}_{ki}(1-\mathcal{A}_{ki})}{\gamma+\mathcal{D}_{kk}}$$

$$\leq \sum_{k\neq i}\frac{\mathcal{A}_{ki}}{\gamma+(n-1)q} \qquad \because \mathcal{D}_{kk}\geq(n-1)q$$

$$= \frac{\mathcal{D}_{ii}}{\gamma+(n-1)q}$$

Therefore

$$\mathbb{E}\left[\left\|\left(\overline{\boldsymbol{S}} - \mathcal{S}\right)\boldsymbol{z}\right\|_2\right] \leq \sqrt{\frac{n}{\gamma+(n-1)q}}$$

**Bound second term in** (22): $\left\|\left(\boldsymbol{S} - \overline{\boldsymbol{S}}\right)\boldsymbol{z}\right\|_2$

Let $\boldsymbol{B} \triangleq \boldsymbol{D}+\gamma\mathbb{I}$ and $\boldsymbol{C} \triangleq \mathcal{D}+\gamma\mathbb{I}$. We first consider the following decomposition:

$$\boldsymbol{B}^{-\frac{1}{2}}\boldsymbol{A}\boldsymbol{B}^{-\frac{1}{2}} - \boldsymbol{C}^{-\frac{1}{2}}\boldsymbol{A}\boldsymbol{C}^{-\frac{1}{2}}$$
$$= \boldsymbol{B}^{-\frac{1}{2}}\boldsymbol{A}\boldsymbol{B}^{-\frac{1}{2}} - \boldsymbol{B}^{-\frac{1}{2}}\boldsymbol{A}\boldsymbol{B}^{-\frac{1}{2}}\boldsymbol{B}^{\frac{1}{2}}\boldsymbol{C}^{-\frac{1}{2}} + \boldsymbol{B}^{-\frac{1}{2}}\boldsymbol{A}\boldsymbol{B}^{-\frac{1}{2}}\boldsymbol{B}^{\frac{1}{2}}\boldsymbol{C}^{-\frac{1}{2}} - \underbrace{\boldsymbol{C}^{-\frac{1}{2}}\boldsymbol{B}^{\frac{1}{2}}\boldsymbol{B}^{-\frac{1}{2}}\boldsymbol{A}\boldsymbol{B}^{-\frac{1}{2}}\boldsymbol{B}^{\frac{1}{2}}\boldsymbol{C}^{-\frac{1}{2}}}_{\text{equal to } \boldsymbol{C}^{-\frac{1}{2}}\boldsymbol{A}\boldsymbol{C}^{-\frac{1}{2}}}$$

$$= \underbrace{\boldsymbol{B}^{-\frac{1}{2}}\boldsymbol{A}\boldsymbol{B}^{-\frac{1}{2}}}_{S}\left(\mathbb{I}-\boldsymbol{B}^{\frac{1}{2}}\boldsymbol{C}^{-\frac{1}{2}}\right) + \left(\mathbb{I}-\boldsymbol{C}^{-\frac{1}{2}}\boldsymbol{B}^{\frac{1}{2}}\right)\underbrace{\boldsymbol{B}^{-\frac{1}{2}}\boldsymbol{A}\boldsymbol{B}^{-\frac{1}{2}}}_{S}\boldsymbol{B}^{\frac{1}{2}}\boldsymbol{C}^{-\frac{1}{2}} \quad (25)$$

Using (25) we can bound the expectation of $\left\|\left(\boldsymbol{S} - \overline{\boldsymbol{S}}\right)\boldsymbol{z}\right\|_2$ as:

$$\mathbb{E}\left[\left\|\left(\boldsymbol{S} - \overline{\boldsymbol{S}}\right)\boldsymbol{z}\right\|_2\right]$$

$$= \mathbb{E}\left[\left(\left(\boldsymbol{D}+\gamma\mathbb{I}\right)^{-\frac{1}{2}}(\boldsymbol{A}+\gamma\mathbb{I})(\boldsymbol{D}+\gamma\mathbb{I})^{-\frac{1}{2}} - (\mathcal{D}+\gamma\mathbb{I})^{-\frac{1}{2}}(\boldsymbol{A}+\gamma\mathbb{I})(\mathcal{D}+\gamma\mathbb{I})^{-\frac{1}{2}}\right)\boldsymbol{z}\right]$$

$$\leq \mathbb{E}\left[\left\|\boldsymbol{S}\left(\mathbb{I}-(\boldsymbol{D}+\gamma\mathbb{I})^{\frac{1}{2}}(\mathcal{D}+\gamma\mathbb{I})^{-\frac{1}{2}}\right)\boldsymbol{z}\right\|_2\right] \quad (26)$$

$$+ \mathbb{E}\left[\left\|\left(\mathbb{I}-(\boldsymbol{D}+\gamma\mathbb{I})^{\frac{1}{2}}(\mathcal{D}+\gamma\mathbb{I})^{-\frac{1}{2}}\right)\boldsymbol{S}(\boldsymbol{D}+\gamma\mathbb{I})^{\frac{1}{2}}(\mathcal{D}+\gamma\mathbb{I})^{-\frac{1}{2}}\boldsymbol{z}\right\|_2\right] \quad (27)$$

Bound (26):

$$\mathbb{E}\left[\left\|\boldsymbol{S}\left(\mathbb{I}-(\boldsymbol{D}+\gamma\mathbb{I})^{\frac{1}{2}}(\mathcal{D}+\gamma\mathbb{I})^{-\frac{1}{2}}\right)\boldsymbol{z}\right\|_2\right] \le \mathbb{E}\left[\|\boldsymbol{S}\|_2\left\|\left(\mathbb{I}-(\boldsymbol{D}+\gamma\mathbb{I})^{\frac{1}{2}}(\mathcal{D}+\gamma\mathbb{I})^{-\frac{1}{2}}\right)\boldsymbol{z}\right\|_2\right]$$

$$\le \sqrt{\sum_i \mathbb{E}\left[\left(1-\sqrt{\frac{\boldsymbol{D}_{ii}+\gamma}{\mathcal{D}_{ii}+\gamma}}\right)^2 \boldsymbol{z}_i^2\right]} \quad \because \|\boldsymbol{S}\|_2 \le 1$$

$$\le \sqrt{\sum_i \mathbb{E}\left[\left(1-\sqrt{\frac{\boldsymbol{D}_{ii}+\gamma}{\mathcal{D}_{ii}+\gamma}}\right)^2\right]}$$

we therefore now need to compute $\sum_i \mathbb{E}\left[\left(1-\sqrt{\frac{\boldsymbol{D}_{ii}+\gamma}{\mathcal{D}_{ii}+\gamma}}\right)^2\right]$. Note that for $x \ge 0$, $|1-\sqrt{x}| \le |1-x|$. Using this we write

$$\sum_i \mathbb{E}\left[\left(1-\sqrt{\frac{\boldsymbol{D}_{ii}+\gamma}{\mathcal{D}_{ii}+\gamma}}\right)^2\right] \le \sum_i \mathbb{E}\left[\left(1-\frac{\boldsymbol{D}_{ii}+\gamma}{\mathcal{D}_{ii}+\gamma}\right)^2\right]$$

$$= \sum_i 1 - 2 + \frac{\mathbb{E}\left[(\boldsymbol{D}_{ii}+\gamma)^2\right]}{(\mathcal{D}_{ii}+\gamma)^2}$$

$$= \sum_i -1 + \frac{\mathbb{E}\left[(\gamma+\sum_{k\ne i}\boldsymbol{A}_{ik})^2\right]}{(\mathcal{D}_{ii}+\gamma)^2}$$

$$= -n + \sum_i \frac{(\mathcal{D}_{ii}+\gamma)^2 + \mathcal{D}_{ii} + \sum_{k\ne i}\boldsymbol{A}_{ik}^2}{(\mathcal{D}_{ii}+\gamma)^2}$$

$$= \sum_i \frac{\sum_{k\ne i}\boldsymbol{A}_{ik}(1-\boldsymbol{A}_{ik})}{(\mathcal{D}_{ii}+\gamma)^2}$$

$$\le \sum_i \frac{1}{\mathcal{D}_{ii}+\gamma}$$

$$\le \frac{n}{\gamma+(n-1)q} \tag{28}$$

Bound (27):

$$\mathbb{E}\left[\left\|\left(\mathbb{I}-(\boldsymbol{D}+\gamma\mathbb{I})^{\frac{1}{2}}(\mathcal{D}+\gamma\mathbb{I})^{-\frac{1}{2}}\right)\boldsymbol{S}(\boldsymbol{D}+\gamma\mathbb{I})^{\frac{1}{2}}(\mathcal{D}+\gamma\mathbb{I})^{-\frac{1}{2}}\boldsymbol{z}\right\|_2\right]$$

$$\le \mathbb{E}\left[\left\|\mathbb{I}-(\boldsymbol{D}+\gamma\mathbb{I})^{\frac{1}{2}}(\mathcal{D}+\gamma\mathbb{I})^{-\frac{1}{2}}\right\|_2\|\boldsymbol{S}\|_2\left\|(\boldsymbol{D}+\gamma\mathbb{I})^{\frac{1}{2}}(\mathcal{D}+\gamma\mathbb{I})^{-\frac{1}{2}}\boldsymbol{z}\right\|_2\right]$$

$$\le \mathbb{E}\left[\max_i\left(1-\sqrt{\frac{(\boldsymbol{D}+\gamma\mathbb{I})_{ii}}{(\mathcal{D}+\gamma\mathbb{I})_{ii}}}\right)\left\|(\boldsymbol{D}+\gamma\mathbb{I})^{\frac{1}{2}}(\mathcal{D}+\gamma\mathbb{I})^{-\frac{1}{2}}\boldsymbol{z}\right\|_2\right]$$

$$\le \left(\underbrace{\mathbb{E}\left[\max_i\left(1-\sqrt{\frac{\boldsymbol{D}_{ii}+\gamma}{\mathcal{D}_{ii}+\gamma}}\right)\right]}_{term1}\underbrace{\mathbb{E}\left[\left\|(\boldsymbol{D}+\gamma\mathbb{I})^{\frac{1}{2}}(\mathcal{D}+\gamma\mathbb{I})^{-\frac{1}{2}}\boldsymbol{z}\right\|^2\right]}_{term2}\right)^{\frac{1}{2}} \tag{29}$$

where (29) follows from applying the Cauchy-Schwarz inequality. Then for (29) *term 2* we get:

$$\mathbb{E}\left[\left\|(\boldsymbol{D}+\gamma\mathbb{I})^{\frac{1}{2}}(\mathcal{D}+\gamma\mathbb{I})^{-\frac{1}{2}}\boldsymbol{z}\right\|^2\right] = \sum_i \mathbb{E}\left[\frac{\boldsymbol{D}_{ii}+\gamma}{\mathcal{D}_{ii}+\gamma}\boldsymbol{z}_i^2\right]$$

$$= \sum_i \underbrace{\frac{\mathbb{E}\left[\boldsymbol{D}_{ii}+\gamma\right]}{\mathcal{D}_{ii}+\gamma}}_{=1}$$

$$= n$$

(29) *term 1* we again note that for $x \geq 0$, $|1-\sqrt{x}| \leq |1-x|$. Using this we write:

$$\mathbb{E}\left[\max_i\left(1-\sqrt{\frac{\boldsymbol{D}_{ii}+\gamma}{\mathcal{D}_{ii}+\gamma}}\right)^2\right] \leq \mathbb{E}\left[\max_i\left(1-\frac{\boldsymbol{D}_{ii}+\gamma}{\mathcal{D}_{ii}+\gamma}\right)^2\right]$$

$$\leq \frac{1}{s}\ln\left(\exp\left(\mathbb{E}\left[s\max_i\left(1-\frac{\boldsymbol{D}_{ii}+\gamma}{\mathcal{D}_{ii}+\gamma}\right)^2\right]\right)\right)$$

$$\leq \frac{1}{s}\ln\left(\mathbb{E}\left[\exp\left(s\max_i\left(1-\frac{\boldsymbol{D}_{ii}+\gamma}{\mathcal{D}_{ii}+\gamma}\right)^2\right)\right]\right)$$

$$= \frac{1}{s}\ln\left(\mathbb{E}\left[\max_i\left(\exp s\left(\left(1-\frac{\boldsymbol{D}_{ii}+\gamma}{\mathcal{D}_{ii}+\gamma}\right)^2\right)\right)\right]\right)$$

$$\leq \frac{1}{s}\ln\left(\sum_i\mathbb{E}\left[\exp\left(s\underbrace{\left(1-\frac{\boldsymbol{D}_{ii}+\gamma}{\mathcal{D}_{ii}+\gamma}\right)^2}_{\boldsymbol{y}_i}\right)\right]\right) \qquad (30)$$

$$\underbrace{\qquad\qquad\qquad\qquad\qquad\qquad\qquad\qquad}_{term1}$$
$$\underbrace{\qquad\qquad\qquad\qquad\qquad\qquad\qquad\qquad\qquad\qquad\qquad}_{term2}$$

Now to further bound (30) we first compute (30), term 1 as:

$$\exp(s\boldsymbol{y}_i) = 1 + s\boldsymbol{y}_i + \sum_{k\geq 2}\frac{(s\boldsymbol{y}_i)^k}{k!}$$

$$= 1 + s\boldsymbol{y}_i + (s\boldsymbol{y}_i)\sum_{k\geq 2}\frac{(s\boldsymbol{y}_i)^{k-1}}{k!}$$

$$= 1 + s\boldsymbol{y}_i + (s\boldsymbol{y}_i)\sum_{k\geq 0}\frac{(s\boldsymbol{y}_i)^k}{(k+1)k!}$$

$$\leq 1 + s\boldsymbol{y}_i + (s\boldsymbol{y}_i)\exp(s\boldsymbol{y}_i)$$

$$\leq 1 + (\exp(s)+1)s\boldsymbol{y}_i$$

Taking the expectation over the previous line, using linearity of expectation and the expression for $\sum_i \mathbb{E}\left[\boldsymbol{y}_i\right]$ from (28) it follows that for (30), term 2 we obtain

$$\sum_i \mathbb{E}\left[\exp(s\boldsymbol{y}_i)\right] \leq n + (\exp(s)+1)s\sum_i \mathbb{E}\left[\boldsymbol{y}_i\right]$$

$$= n + (\exp(s)+1)s\frac{n}{\gamma+(n-1)q}$$

Going back to (30):

$$(30) \le \frac{1}{s} \ln \left( n + (\exp(s) + 1)s \frac{n}{\gamma + (n-1)q} \right) \qquad \forall s > 0$$

$$\le \frac{1}{s} \ln \left( n + \exp(2s) \frac{n}{\gamma + (n-1)q} \right) \qquad \text{Note: } s > 0 \Rightarrow \ln s \le s - 1$$

$$\Rightarrow (e^s + 1)s \le e^{2s}$$

$$\le \frac{\ln(n)}{s} + \frac{1}{s} \ln \left( 1 + \frac{\exp(2s)}{\gamma + (n-1)q} \right) \qquad \text{Let } e^{2s} \ge \gamma + (n-1)q$$

$$\le \frac{\ln(n)}{s} + \frac{1}{s} \ln \left( \frac{2 \exp(2s)}{\gamma + (n-1)q} \right)$$

$$\le \frac{\ln(n)}{s} + 2 + \frac{1}{s} \ln \left( \frac{2}{\gamma + (n-1)q} \right) \qquad \text{Take } s := \gamma + (n-1)q \ge 2$$

$$\le C \frac{\ln(n)}{\gamma + (n-1)q}$$

Finally combining the above results:

$$\mathbb{E}\left[ \left\| (\boldsymbol{S} - \overline{\boldsymbol{S}}) \boldsymbol{z} \right\|_2 \right] \le \sqrt{\frac{n}{\gamma + (n-1)q}} + \sqrt{n \frac{C \ln(n)}{\gamma + (n-1)q}}$$

$$= C \sqrt{\frac{n \ln(n)}{\gamma + (n-1)q}}$$

and

$$\mathbb{E}\left[ \left\| (\boldsymbol{S} - \mathcal{S}) \mathcal{X} \right\|_{2 \to \infty} \right] \le C \sqrt{\frac{n \ln n}{\gamma + (n-1)q}} \, \|\boldsymbol{\mu}\|_\infty$$

This concludes he bound of $\mathbb{E}\left[ \left\| (\boldsymbol{S} - \mathcal{S}) \mathcal{X} \right\|_{2 \to \infty} \right]$. $\qquad\qquad \square$

### E.3.2 Bound $\mathbb{E}\left[ \left\| (\boldsymbol{X} - \mathcal{X}) \boldsymbol{S} \right\|_{2 \to \infty} \right]$

We first note that

$$\mathbb{E}\left[ \left\| (\boldsymbol{X} - \mathcal{X}) \boldsymbol{S} \right\|_{2 \to \infty} \right] = \mathbb{E}\left[ \max_{j \in [d]} \|\boldsymbol{S} \epsilon_{\cdot j}\|_2 \right]$$

$$\le \left( \mathbb{E}\left[ \max_{j \in [d]} \|\boldsymbol{S} \epsilon_{\cdot j}\|_2^2 \right] \right)^{\frac{1}{2}}$$

Let $z \sim \mathcal{N}(0, \sigma^2 \mathbb{I})$ then

$$\|\boldsymbol{S} \boldsymbol{z}\|_2^2 = \boldsymbol{z}^\top \boldsymbol{S}^\top \boldsymbol{S} \boldsymbol{z}$$

$$= \boldsymbol{z} \boldsymbol{V} \boldsymbol{\Lambda} \boldsymbol{V}^\top \boldsymbol{z} \qquad \text{Eigendecompsition}$$

$$= \sum_{i=1}^{n} \lambda_i z_i'^2 \qquad \text{where } \boldsymbol{V}^\top \boldsymbol{z} = \boldsymbol{z}_i' \sim \mathcal{N}(0, \sigma^2 \mathbb{I})$$

$$= \sum_{i=1; \lambda_i > 0}^{n} \lambda_i \sigma^2 \boldsymbol{y}_i \qquad \boldsymbol{y}_i, \cdots, \boldsymbol{y}_d \overset{iid}{\sim} \mathcal{X}^2$$

Where the first line follows from the eigendecomposition $\boldsymbol{S}^\top \boldsymbol{S} = \boldsymbol{V} \boldsymbol{\Lambda} \boldsymbol{V}^\top$. Therefore $\|\boldsymbol{S} \boldsymbol{z}\|_2^2$ is distributed as a generalised $\mathcal{X}^2$ with mean $\sigma \operatorname{Tr}(\boldsymbol{S}^\top \boldsymbol{S})$ and variance $2 \sum \lambda_i \sigma^4 = 2 \sigma^4 \left\| \boldsymbol{S}^\top \boldsymbol{S} \right\|_F^2$.
Now define

$$\mathrm{MGF}_y(s) = \frac{1}{\exp \left( \frac{1}{2} \sum_{i: \lambda_i > 0} \log(1 - 2s\lambda_i) \right)}$$

and consider $s \in \left(0, \frac{1}{2\lambda_{min}}\right)$ where $\lambda_{min}$ is the smallest non-zero eigenvalue of $\boldsymbol{S}^\top \boldsymbol{S}$.

$$\exp\left(s\mathbb{E}\left[\max_j \boldsymbol{y}_j\right]\right) \leq \mathbb{E}\left[\exp\left(s\max(\boldsymbol{y}_j)\right)\right]$$
$$= \mathbb{E}\left[\max \exp\left(s\boldsymbol{y}_j\right)\right]$$
$$\leq \sum_j \mathbb{E}\left[\exp\left(s\boldsymbol{y}_j\right)\right]$$
$$= d \cdot \mathrm{MGF}_{\boldsymbol{y}}(s)$$
$$= d\exp\left(-\frac{1}{2}\sum_{i:\lambda_i>0} \log(1 - 2s\lambda_i)\right)$$

it follows that

$$\mathbb{E}\left[\max_j \boldsymbol{y}_j\right] \leq \frac{\ln d}{s} - \frac{1}{2s}\sum_{i:\lambda_i>0} \underbrace{\log(1 - 2s\lambda_i)}_{\leq -2s\lambda_i}$$
$$\leq \frac{\ln d}{s} + \underbrace{\sum_{i:\lambda_i>0} \lambda_i}_{\mathrm{Tr}(\boldsymbol{S}^\top \boldsymbol{S})} \qquad\qquad \because \log(1+x) \leq x \ \forall x > -1$$
$$\leq 2\lambda_{min}\ln d + \mathrm{Tr}(\boldsymbol{S}^\top \boldsymbol{S}) \qquad \because s \in \left(0, \frac{1}{2\lambda_{min}}\right) \text{ and min for } s = \frac{1}{2\lambda_{min}}$$

Using $\sigma_{min}(\boldsymbol{S}) \leq \|\boldsymbol{S}\|_2$ and $\|\boldsymbol{S}\|_F \leq k\|\boldsymbol{S}\|_2$ we can bound the last line as $\|\boldsymbol{S}\|_2^2 (k + 2\ln d)$ in the low-rank setting. However since we consider $\boldsymbol{S}$ to be random this is not applicable (also see the remarks in the VC Dimension section). Therefore

$$2\lambda_{min}\ln d + \mathrm{Tr}(\boldsymbol{S}^\top \boldsymbol{S}) = \sigma_{min}^2(\boldsymbol{S})\ln d + \|\boldsymbol{S}\|_F^2$$
$$\leq \|\boldsymbol{S}\|_F^2 (1 + 2\ln d)$$

and taking the square root gives us the final result:

$$\mathbb{E}\left[\|(\boldsymbol{X} - \mathcal{X})\boldsymbol{S}\|_{2\to\infty}\right] \leq \sigma\|\boldsymbol{S}\|_F \sqrt{1 + 2\ln d}$$

**Bound $\mathbb{E}\left[\|\boldsymbol{S}\|_F^2\right]$.**

Case 1: Self loop.

We first note that $\|\boldsymbol{S}\|_F^2 = n + \textit{number of edges}$ and therefore:

$$\mathbb{E}\left[\|\boldsymbol{S}\|_F^2\right] \leq n + n^2 p$$
$$= (1 + o(1))n^2 p$$

Therefore

$$\mathbb{E}\left[\|(\boldsymbol{X} - \mathcal{X})\boldsymbol{S}\|_{2\to\infty}^2\right] \leq (1 + o(1))n^2 p\sigma^2(1 + 2\ln d)$$

Case 2: Degree normalized.

Note that we here overload the notation $d$ such that we define the degree for node $i$ as $d_i$ and similar $d_{min}$ is the minimum degree.

$$\mathbb{E}\left[\|\boldsymbol{S}\|_F^2\right] = \mathbb{E}\left[\|\boldsymbol{S}\|_F^2\Big|\left\{d_{min} > np - \sqrt{4cnp\ln n}\right\}\right]\mathbb{P}\left(d_{min} > np - \sqrt{4cnp\ln n}\right)$$
$$+ \mathbb{E}\left[\|\boldsymbol{S}\|_F^2\Big|\left\{d_{min} < np - \sqrt{4cnp\ln n}\right\}\right]\mathbb{P}\left(d_{min} < np - \sqrt{4cnp\ln n}\right)$$
$$\leq \mathbb{E}\left[\|\boldsymbol{S}\|_F^2\Big|\left\{d_{min} > np - \sqrt{4cnp\ln n}\right\}\right]\mathbb{P}\left(d_{min} > np - \sqrt{4cnp\ln n}\right) + \underbrace{n^2\frac{1}{n^c}}_{=o(1)}$$

$$\leq \sum_{i,j}\frac{\boldsymbol{A}_{ij} + \mathbb{I}\{i = j\}}{(d_i + 1)(d_j + 1)}$$

$$\leq \frac{1}{d_{min} + 1}\sum_i \underbrace{\frac{\sum_j \boldsymbol{A}_{ij} + \mathbb{I}\{i = j\}}{d_i + 1}}_{=1}$$

$$\leq \frac{n}{nq + 1 - \sqrt{4cnp\ln n}}$$

$$= (1 + o(1))\frac{1}{q}$$

Therefore

$$\mathbb{E}\left[\|(\boldsymbol{X} - \mathcal{X})\,\boldsymbol{S}\|_{2\to\infty}^2\right] \leq (1 + o(1))\frac{\sigma^2(1 + 2\ln d)}{q}$$

This concludes the bound of $\mathbb{E}\left[\|(\boldsymbol{X} - \mathcal{X})\,\boldsymbol{S}\|_{2\to\infty}^2\right]$. $\qquad\square$

### E.3.3 Bound $\mathbb{E}\left[\|\boldsymbol{S}\|_\infty^k\right]$.

In general we can note that $\|\mathcal{S}\|_\infty^k = \max_{1\leq i\leq n}\left(\sum_{j=1}^n \mathcal{S}_{ij}\right)^k$

Case 1: Self loop.

We first define the degree for node $i$ as

$$d_i \sim \text{Bin}\left(\frac{n}{2} - 1, p\right) + \text{Bin}\left(\frac{n}{2}, q\right)$$

then $\|\mathcal{S}\|_\infty = \max_{1\leq i\leq n}\left(\sum_{j=1}^n \mathcal{S}_{ij}\right) = 1 + \max_i d_i$ and assume $p > \frac{\ln n}{n}$ and let $t = \sqrt{4np\ln n}$

$$\mathbb{P}\left(d_i - \mathbb{E}\left[d_i\right] > t\right) \leq \exp\left(\frac{-\frac{t^2}{2}}{np + \frac{t}{3}}\right) \qquad \text{Bernstein inequality}$$

$$\leq \exp\left(\frac{-4cnp\ln n}{4np}\right)$$

$$= \frac{1}{n}c$$

and therefore

$$\mathbb{P}\left(\max_i d_i \geq np + \sqrt{4cnp\ln n}\right) \leq \frac{1}{n^{c-1}}$$

$$\mathbb{P}\left((1 + \max_i d_i)^k \geq (1 + np + \sqrt{4cnp\ln n})^k\right) \leq \frac{1}{n}c$$

and

$$\mathbb{E}\left[(1 + \max_i d_i)^k\right] \leq (1 + np + \sqrt{4cnp\ln n})^k + \frac{1}{n^{c-i}}n^k$$

$$= (1 + np + \sqrt{4cnp\ln n})^k + n^{k+1-c}$$

For large $n$ and $p \gg \frac{(\ln n)^2}{n}$ take $c = \ln n$:

$$\mathbb{E}\left[\|\boldsymbol{S}\|_\infty^k\right] \leq ((1 + o(1))np)^k$$

Case 2: Degree normalized.

$$\|\boldsymbol{S}\|_\infty = \max_i \sum_j \boldsymbol{S}_{ij}$$

$$= \max_i \sum_j \frac{\boldsymbol{A}_{ij}}{\sqrt{d_i + 1}\sqrt{d_j + 1}}$$

$$\leq \max_i \frac{1}{\sqrt{d_{min} + 1}} \frac{\sum_j \boldsymbol{A}_{ij}}{\sqrt{d_i + 1}}$$

$$= \max_i \sqrt{\frac{d_i + 1}{d_{min} + 1}}$$

$$\leq \sqrt{\frac{d_min + 1}{d_{min} + 1}}$$

Similar to above we can now note that:

$$\mathbb{P}\left(\max_i d_i \geq np + \sqrt{4cnp\ln n}\right) \leq \frac{1}{n^c}$$

$$\mathbb{P}\left(\max_i d_i \leq np + \sqrt{4cnp\ln n}\right) \leq \frac{1}{n^c}$$

and it follows

$$\mathbb{P}\left(\sqrt{\frac{d_{max} + 1}{d_{min} + 1}} \geq \frac{np + \sqrt{4cnp\ln n} + 1}{np - \sqrt{4cnp\ln n} + 1}\right) \leq \frac{2}{n^c}$$

For large $n$ and $p, q \gg \frac{(\ln n)^2}{n}$:

$$\mathbb{E}\left[\|\boldsymbol{S}\|_\infty^k\right] \leq \mathbb{E}\left[\left(\frac{d_{max} + 1}{d_{min} + 1}\right)^{\frac{k}{2}}\right]$$

$$= \left((1 + o(1))\frac{p}{q}\right)^{\frac{k}{2}}$$

This concludes the bound of $\mathbb{E}\left[\|\boldsymbol{S}\|_\infty^k\right]$. $\qquad\square$

### E.3.4 Bound $\|\mathcal{S}\mathcal{X}\|_{2\to\infty}$.

Case 1: Self loop.

$$\mathcal{S}\mathcal{X} = (1 - p)\boldsymbol{z}\boldsymbol{\mu}^\top - \frac{p - q}{2}\boldsymbol{y}\boldsymbol{y}^\top \boldsymbol{z}\boldsymbol{\mu}^\top$$

$$= \left((1 - p)\boldsymbol{z} - \left(\frac{p - q}{2}\boldsymbol{y}^\top \boldsymbol{z}\right)\boldsymbol{y}\right)\boldsymbol{\mu}^\top$$

and

$$(\mathcal{SX})_{ij} = \left( (1-p)\boldsymbol{z}_i - \underbrace{\left( \frac{p-q}{2} \boldsymbol{y}^\top \boldsymbol{z} \right) \boldsymbol{y}_i}_{\triangleq \delta} \right) \boldsymbol{\mu}_j$$

Now using this to compute the two-infinity norm:

$$
\begin{aligned}
\|\mathcal{SX}\|_{2\to\infty} &= \|\boldsymbol{\mu}\|_\infty \sqrt{\sum_i ((1-p)\boldsymbol{z}_i - \delta \boldsymbol{y}_i)^2} \\
&= \|\boldsymbol{\mu}\|_\infty \sqrt{\sum_i (1-p)^2 + \delta^2 - 2\delta(1-p)\boldsymbol{y}_i \boldsymbol{z}_i} \\
&= \|\boldsymbol{\mu}\|_\infty \left( n(1-p)^2 + n(\boldsymbol{y}^\top \boldsymbol{z})^2 \left( \frac{p-q}{2} \right)^2 - 2(\boldsymbol{y}^\top \boldsymbol{z})^2 \frac{p-q}{2}(1-p) \right) \\
&= (1+o(1)) \|\boldsymbol{\mu}\|_\infty n \left( 1 + \left( \frac{p-q}{2} \right)^2 (\boldsymbol{y}^\top \boldsymbol{z})^2 \right)
\end{aligned}
$$

Case 2: Degree normalized.

We note that the expected degree is $(1+o(1))n\frac{p+q}{2}$ and therefore similar to above we obtain

$$\|\mathcal{SX}\|_{2\to\infty} = (1+o(1)) \|\boldsymbol{\mu}\|_\infty \frac{\left( 1 + \left( \frac{p-q}{2} \right)^2 (\boldsymbol{y}^\top \boldsymbol{z})^2 \right)}{\left( \frac{p+q}{2} \right)}.$$

This concludes the bound of $\|\mathcal{SX}\|_{2\to\infty}$. $\qquad\square$

# F Experimental Details

## F.1 Data

**SBM.** For the SBM experiments we follow the description in the main paper: assume that the node features are sampled latent true classes, given a $\boldsymbol{z} = (z_1, \ldots, z_n) \in \{\pm 1\}^n$. The node features are sampled from a Gaussian mixture model (GMM), that is, feature for node-$i$ is sampled as $\boldsymbol{x}_i \sim \mathcal{N}(z_i \boldsymbol{\mu}, \sigma^2 \mathbb{I})$ for some $\boldsymbol{\mu} \in \mathbb{R}^d$ and $\sigma \in (0, \infty)$. We express this in terms of $\boldsymbol{X}$ as

$$\boldsymbol{X} = \mathcal{X} + \boldsymbol{\epsilon} \in \mathbb{R}^{n \times d}, \qquad \text{where } \mathcal{X} = \boldsymbol{z} \boldsymbol{\mu}^\top \text{ and } \boldsymbol{\epsilon} = (\epsilon_{ij})_{i \in [n], j \in [d]} \overset{i.i.d.}{\sim} \mathcal{N}(0, \sigma^2).$$

We refer to above as $\boldsymbol{X} \sim 2\text{GMM}$. On the other hand, we assume that graph has two latent communities, characterised by $\boldsymbol{y} \in \{\pm 1\}^n$. The graph is generated from a stochastic block model with two classes (2SBM), where edges $(i, j)$ are added independently with probability $p \in (0, 1]$ if $y_i = y_j$, and with probability $q < [0, p)$ if $y_i \neq y_j$. In other words, we define the random adjacency $\boldsymbol{A} \sim 2\text{SBM}$ as a symmetric binary matrix with $\boldsymbol{A}_{ii} = 0$, and $(\boldsymbol{A}_{ij})_{i<j}$ indenpendent such that

$$\boldsymbol{A}_{ij} \sim \text{Bernoulli}(\mathcal{A}_{ij}), \qquad \text{where } \mathcal{A} = \frac{p+q}{2} \boldsymbol{1}\boldsymbol{1}^\top + \frac{p-q}{2} \boldsymbol{y}\boldsymbol{y}^\top - p\mathbb{I}.$$

The choice of two different latent classes $\boldsymbol{z}, \boldsymbol{y} \in \{\pm 1\}^n$ allows study of the case where the graph and feature information of do not align completely.

Therefore for to characterise the model we need to define: $p, q, n, \boldsymbol{z}, \boldsymbol{y}, \boldsymbol{\mu}, \sigma$

**Cora.** For the real world experiments we use the cora dataset Rossi et al. (2015)[4]. The dataset consists of 2708 machine learning papers and is split into seven classes: *Case_Based, Genetic_Algorithms, Neural_Networks, Probabilistic_Methods, Reinforcement_Learning, Rule_Learning, Theory*. The features are a bag of words of size 1433.

## F.2 Experiments Section 3.2

### F.2.1 SBM

**Setup and Data.** We consider the synthetic data to be generated as defined in (6) and (7). We sample the SBM with the following parameters as default: $n = 500, d = 100, p = 0.2, q = 0.01, \Gamma = n, m = 100, u = 400$. $\boldsymbol{\mu}$ is sampled uniformly. The GNN is by default a one layer model $K = 1$ with hidden layer size $d_1 = 16$, ReLu activation, $\phi(\cdot) = \text{ReLU}(\cdot)$ and squared loss. Plotted is the error over the displayed change of parameters for epochs[5] between 50 and 1000 (over 50 intervals). We plot the results averaged over five random initialisation.

**Change alignment.** We consider the SBM[6] setting as defined above while now varying $\Gamma \in (0, n)$ over 10 steps and for easier readability plot $\frac{\Gamma}{n}$. The GNN is optimized using SGD with learning rate 0.001.

**Change graph size.** We consider the SBM setting as defined above with setting again $\frac{\Gamma}{n} = 1$ while now varying the graph size $n \in (200, 2000)$ over 10 steps while adjusting $\frac{m}{n}$ accordingly. The GNN is optimized using SGD with learning rate 0.01.

**Change number of marked points.** We consider the SBM setting as defined above with $\frac{\Gamma}{n} = 0.7, p = 0.2, q = 0.15$ while now varying the number of observe points such that $\frac{m}{n} \in (0.01, 0.05)$ over 10 steps. The GNN is optimized using SGD with learning rate 0.2.

**Plot theoretical bound.** Recall that for plotting the theoretical bound we can only plot the trend of the bound as the absolute value is out of the $(0, 1)$ range. This problem is inherent to the bound given in El-Yaniv et al. (2009) that we base our TRC bounds on, as the slack terms can already exceeds 1

---

[4]Using the import from `https://github.com/tkipf/pygcn/tree/master/data/cora`

[5]A consideration of different epochs is important as the presented bounds do not take the optimization explicitly into consideration. As stated previously a future way to do so could be by analysing the behaviour of the the bounds on the parameters during optimization.

[6]Remark on change in training and SBM setting: Since we are interested in upper bounds we observe that under some settings the trends are more clear then in others. For example for some learning rate the change might be less obvious then for the reported one.

and therefore further research on general TRC generalisation gaps is necessary to characterise the absolute gap between theory and experiments. More specifically we scale *SBM, change alignment* and *SBM, change graph size* by a factor of 25 and *SBM, change number of marked points* by a factor of 30. Again as noted in the main paper we fix the bounds on the on the learnable parameters for plotting the theoretical bounds. From samples we observe that $\beta, \omega \approx 0.1$ and therefore consider this for the plots. A more detailed analysis of this will be necessary in future research to investigate how the change of those bounds changes the generalisation error bound.

### F.2.2 Cora

**Setup and Data.** We now consider the *Cora* dataset with $n = 2708$ and $\frac{m}{n} = 0.1$. The GNN follows the setup of the SBM with the difference that we now consider a multi-class problem. Therefore a *negative log likelihood loss* is considered. In addition we consider the Adam optimizer Kingma et al. (2015) with learning rate 0.01.

**Change alignment.** We simulate a change in the feature structure by adding noise to the feature vector as $\boldsymbol{X} + \epsilon$ where $\epsilon_i$. is *i.i.d.* distributed $\mathcal{N}(0, \sigma_{\text{Feat}}^2 \mathbb{I})$ and again observe a similar behaviour to the SBM. We vary $\sigma_{\text{Feat}} \in (0, 0.1)$ over 10 steps.

**Cora, change graph size.** To change the graph size we sample 10 sub-graphs of size $n \in (1354, 2708)$.

**Change number of marked points.** For varying the number of observe points we consider $\frac{m}{n} \in (0.05, 0.3)$ over 10 steps.

### F.3 Experiments Section 4.2 (Residual connections)

**Setup and Data.** We consider the same general setup as above (section F.2). We now change the parameter $K$. For implementing residual connections we slightly deviate from (13) by considering the residual connection to be to the first layer instead of the features directly. This change follows Chen et al. (2020) where the residual connection was proposed as otherwise the size of the hidden layer would be fixed to $n$. For the experiments we consider $d_i = 16 \ \forall i \in [K]$.

**Change depth.** For both datasets we now changed the depth for $K \in [4]$ and two different residual connections with $\alpha \in \{0.2, 0.5\}$.

### F.4 Implementation

For the implementation of the GNN we use official code of Kipf et al. (2017)[7] as a foundation that is provided under an *MIT License*.

Experiments are ran on a MacBook Pro (16-inch, 2019), processor 2,3 GHz 8-Core Intel Core i9, memory 32 GB 2667 MHz DDR4.

---

[7] https://github.com/tkipf/pyGNN