# OpenReview forum: "Learning Theory Can (Sometimes) Explain Generalisation in Graph Neural Networks"
_NeurIPS.cc/2021/Conference — NeurIPS 2021 Poster_

### Official Review · Reviewer_Mq8Q · 2021-07-12

**Rating:** 6
**Confidence:** 3

**Summary:**

The aim of the paper is to provide generalization bounds for graph neural networks (GNNs) in the transductive setting. Several results are presented where the deviation between the expected loss on unlabeled data and empirical loss on labeled data is bounded using the Rademacher complexity of GNN hypothesis classes.

**Limitations And Societal Impact:**

The limitations of the work could be discussed in more detail.

**Main Review:**

The paper is well-written and rigorous. While the presented results seem helpful towards obtaining a better understanding of the performance of GNNs, the authors may find it useful to discuss the following points more clearly:

1. My major concern about the results in this paper is about the bound in Theorem 1.

a. The result in Theorem 1 (and the following results) suggest that the generalization error should increase with the number of unlabeled samples n, due to the terms of O(n^2/(m(n-m))) and O(sqrt(log(n))) in the Rademacher complexity expression. However, none of the experimental results indicate an increase in the generalization error as n increases. I'm wondering whether the increase in the error with n as predicted by the theorems is a side effect of the proof techniques used in the paper, or there may potentially be certain graph topologies where such an increase can indeed be experimentally observed.

b. In the bound in Theorem 1 and the following bounds, the error is seen to include a term of O(n^2/(m(n-m))). Regarding the distribution of labeled vs. unlabeled samples, this result implies that the error takes its smallest value when half of the samples are labeled (m=0.5n). This seems surprising, as one would expect the generalization error to be smaller if more samples are labeled. Is there an intuitive interpretation of this?

2. Theorem 2 suggests that the Rademacher complexity tends to increase with Gamma, which is also supported by the experimental results. Given that Gamma represents the alignment between the graph and the feature information, could the authors provide some interpretation about why the complexity of the GNN hypothesis family is higher when the information provided by the graph topology and the graph features are more aligned? I would intuitively expect the hypothesis function complexity to be smaller when the "intrinsic dimensionality" of the data space is smaller.

3. Corollary 1 and the results in Figure 2 suggest that as alpha increases, the generalization error decreases. However, considering the balance between the feature information propagating from the previous layers and the residual feature information coming from the first layer, one would normally expect the optimal value of alpha to be an intermediate value between 0 and 1 (i.e. one would not expect to obtain the best results at alpha=1). Some additional explanations would be useful here.


4. Minor comments:
- In the bound in Proposition 1, what does the parameter "em" mean?
- I recommend that the authors proofread their paper to check for typos. There are several typos in the paper (e.g. "Randemacher complexity" in Theorem 1, "In the broader deep learning learning" in page 2, and several others).

**Time Spent Reviewing:**

5-6 hours

---

> ### Author Response · Authors · 2021-08-09
> **Response to reviewer’s comments**
>
> We thank the reviewer for the recommendations for the paper, and also for the constructive feedback.
>
> In the following, we address open questions posed in the review. For both points 1b) and 3), we will add additional remarks on the range of values where the bounds follow the general intuition of the behavior to the revised version.
>
> 1a) This effect seems to be mostly due to proof techniques as in all performed experiments (also such that were not included in the paper) displayed a constant behavior over the change of $n$.
>
> 1b) Indeed, this behavior is surprising and also most likely not what would be observed in real-world applications. Unfortunately, this setup seems to be an effect of the proof techniques in [EYP09], as similar quantities are present in [EYP09]. However, in the interesting setting, $m\ll n-m$, where only a few points are observed, the behavior of the bounds seems to be intuitive and aligned with experimental results.
>
> 2) Our interpretation is that with strong alignment, we might observe an overfitting effect that, in this case, seems to be stronger than the effect between the hypothesis function complexity and “intrinsic dimensionality” of the data space as pointed out by the reviewer.
>
> 3) Our general intuition behind this behavior is that with increasing $\alpha$, the network architecture is closer to the one of an one hidden layer network. Having good performance in shallow networks is something that is observed in our experiments as well as in previous work (e.g., [KW17]). Therefore it appears that using the skip connection to obtain a deep network that resembles a shallow one leads to the performance increase. However, we agree that in practical applications, an $\alpha$ close but not equal to one would be more realistic.
>
> 4) Minor comments: The quantity $em$ is part of the general generalization error bound based on VD Dimension and, more specifically, comes from applying Sauer’s Lemma in the process (see for example Lemma 6.10 (Sauer-Shelah-Perles) [SSBD14]). Therefore the quantity does not have an interpretation specifically related to this bound.
>
> Other textual/minor comments we will address in the revised version.
>
> [EYP09] Ran El-Yaniv and Dmitry Pechyony. “Transductive Rademacher Complexity and its Applications”. In: Journal of Artificial Intelligence Research. 2009.
>
> [KW17] Thomas N. Kipf and Max Welling. “Semi-Supervised Classification with Graph Convolutional Networks”. In: International Conference on Learning Representations. 2017.
>
> [SSBD14] Shai Shalev-Shwartz and Shai Ben-David. Understanding Machine Learning: From Theory to Algorithms. Cambridge University Press, 2014.

---

> > ### Comment · Reviewer_Mq8Q · 2021-09-02
> > **Reply**
> >
> > I would like to thank the authors for their explanations. I still have a positive opinion about this paper.

---

### Official Review · Reviewer_n6fM · 2021-07-16

**Rating:** 6
**Confidence:** 3

**Summary:**

The problem addressed in this paper is an important area in theoretical understanding graph convolution networks. This paper discusses the transductive learning using the Rademacher complexity analysis with VC-dimensions and data dependent bounds. Further, the paper analyzes the special setting of the GCNII which is known to be robust against oversmoothing. Overall, a good paper, however, some of the interpretations of bounds are not clear, which require further clarifications.

**Limitations And Societal Impact:**

Lack of usage of spectral analysis could be a limitation in interpretations of oversmoothing.

**Main Review:**

The study of generalization bounds for GCN is an important problem which has been recently studied by several researchers. This paper further adds insights to the study of generalization bounds using tranductive Rademacher complexity, VC dimensions, and residual networks.

The study of transductvie Rademacher complexity for GCN has also been studied under restricted conditions in  Kenta Oono, and Taiji Suzuki: Optimization and Generalization Analysis of Transduction through Gradient Boosting and Application to Multi-scale Graph Neural Networks (NeurIPS 2020). This reference is missing in the paper. It would be helpful to have some insight into how the  bounds given in the above paper are related to the proposed work.

In general, the paper is written well. However, there are several typos.
It is very useful to have a notations section to introduce mathematical notations. Otherwise, it is difficult to follow notations without searching through the paper.
In line 684: Should dual be  \| \sigma S X\|_1 ?
Line 797: Line before the eq. 30, is it  s1?
Line 840 d_m in ?

Oversmoothing is an important aspect of GCN model that has gained theoretical attention. This paper also addresses this problem with the setting of GCNII. It is not convincing to use depth of 4 to explain the effect of oversmoothing in figure 2. It would be more clear if larger depths are considered as in [Che+20] where they considered depth from 2,4,...,64.
Further, it is not clear about the interpretation of the effect of oversmoothing from the bounds for plain GCN or the GCNII [Che+20] setting since the theory and experiments disagree as depth increases. Can authors elaborate possible improvements to bounds that can be made to obtain better interpretation in oversmoothing?

Spectral analysis has been employed to explain oversmoothing in Kenta Oono and Taiji Suzuki: Graph Neural Networks Exponentially Lose Expressive Power for Node Classification, ICLR2020. In their paper, it has been shown that as the number of layers increases higher orders of eigenvalues quickly decrease to zeros leading to information losses. Can the generalization bounds be expressed with spectral components (e.g. eigenvalues)?

Finally, it is nice if the inductive setting can also be included in the paper. Is it possible to extend generalization bounds in the paper for the inductive setting?

**Time Spent Reviewing:**

6

---

> ### Author Response · Authors · 2021-08-09
> **Response to reviewer’s comments**
>
> We thank the reviewer for the recommendations for the paper and also for the constructive feedback.
>
> In the following, we address open questions posed in the review.
>
> 1) Relation to [OS20a] (see also [N3] by Reviewer d7vj). We thank the reviewer for pointing out the reference, and we will address it for the revised version. The main differences can be seen as follows:
>
>   * The first main difference is in the setting. [OS20a] considers a multiscale GCN, and therefore, the analysis is based in a weak-learning/boosting framework where the focus is mostly on exploring the weak learning component, whereas this paper focuses on the specific analysis of the generalization bound and the influence of it’s individual components.
>
>   * While [OS20a] includes an expression dependent on $\boldsymbol{S} \boldsymbol{X}$ there is no further analysis provided on this expression, whereas we analyze aspects like normalization from this quantity (eq. 10 and eq. 11).
>
>   * For a more expressive bound, we consider generalization using TRC under Planted Models. An analysis under distributional assumptions is not considered in [OS20a]. However, this is an important step as it allows us to make statements about the influence of graph/feature alignment, the number of observed points, and the influence of graph information.
>
>   * Finally, in contrast to [OS20a], our analysis easily extends to other network architectures, and we validate the observations empirically.
>
> 2) Spectral analysis and oversmoothing. We can express the generalization bound with spectral components as follows. Recall Theorem 1 (Generalization error bound for GNNs using TRC) where the dependency is given with $||\boldsymbol{S}|| _ {\infty}$ and $||\boldsymbol{S} \boldsymbol{X}||_{2 \rightarrow \infty}$. We can further bound both expressions in terms of the spectral norm as follows:
>
> * $\frac{1}{\sqrt{n}}||\boldsymbol{S}|| _ {\infty} \leq||\boldsymbol{S}|| _ {2}$
>
> * $||\boldsymbol{S} \boldsymbol{X}|| _ {2 \rightarrow \infty} = \max _ j||(\boldsymbol{S} \boldsymbol{X}) _ {\cdot j}|| _ 2\leq\max _ j||\boldsymbol{S}|| _ {2}||\boldsymbol{X} _ {\cdot j}|| _ 2\leq||\boldsymbol{S}|| _ {2}|| \boldsymbol{X}|| _ {2 \rightarrow \infty}$
>
> One can connect this to the oversmoothing effect by noting that the diffusion operator in the spectral setting is included as $||\boldsymbol{S}|| _ {2}^k$ and $||\boldsymbol{S}|| _ {2}^K$ for $k\in[K]$ layers. Therefore with an increasing number of layers (and especially in the setting considered in [OS20b] where the number of layers goes to infinity), the information provided by the graph gets oversmoothed and therefore, a loss of information can be observed. This would therefore be in line with the main message in [OS20b].
>
> Note that this provides a weaker bound than the one presented in our paper. Nevertheless, we will add a section on the spectral perspective and over smoothing to the revised version. We shortly touch on this aspect in the end of page 5 under 'From spectral radius to $||\boldsymbol{S} \boldsymbol{X}||_{2 \rightarrow \infty}$'.
>
> 3) Inductive setting. Some proof techniques can be adapted, but the models would end up being different since now there would be different SBMs/graph models - one of each graph/each class. We will add a note to the revised version and consider it as part of future work.
>
> Other textual/minor comments will be addressed in the revised version.
>
> [OS20a] Kenta Oono, and Taiji Suzuki: Optimization and Generalization Analysis of Transduction through Gradient Boosting and Application to Multi-scale Graph Neural Networks (NeurIPS 2020)
>
> [OS20b] Kenta Oono and Taiji Suzuki: Graph Neural Networks Exponentially Lose Expressive Power for Node Classification, ICLR2020

---

> > ### Comment · Reviewer_n6fM · 2021-08-31
> > **Reply**
> >
> > Thank you for the explanations.
> >
> > I think discussions provided in relation to [OS20a] and [OS20b] should be included in the revised version of the paper (at least briefly).  Further, a notation section should be included to summarize the notation to improve the readability.

---

### Official Review · Reviewer_d7vj · 2021-07-16

**Rating:** 5
**Confidence:** 3

**Summary:**

This paper analyzed the generalization performance of GNNs based on statistical learning theory. First, this paper derived the VC dimension-based generalization error bounds. It showed that graph structures affected the upper bound of the generalization performance gap via the rank of the adjacency matrix (and its normalized version). This paper also argued that these gaps are vacuous for some random graphs. For a more precise evaluation, this paper next derived generalization error bounds based on the notion of Transductive Rademacher Complexity (TRC). The effect of the normalization of the adjacency matrix S on the generalization bounds is explained for the general and planted model case. Finally, this paper derived the generalization bounds based on TRC for GNNs with residual connections. It argued that residual connections reduced the generalization gap.

**Ethical Concerns:**

N.A.

**Limitations And Societal Impact:**

- [L1] This paper discussed the limitation of the analysis in terms of model architectures (e.g., not applicable to the models with dropout or batch normalization)


**Main Review:**

### Soundness (Do theorems and experiments answer research questions, assuming they are correct?)

- [So1] The research question of this paper is to analyze the generalization performance of GNNs in a transductive problem setting (l.43) from the viewpoint of statistical learning theory. This paper claimed that although classical learning theory cannot explain the behavior of deep learning models for FNNs, while it can, to some extent, for GNNs. I think this paper certainly answers this question for the following reasons.
  - [So1.1] This paper analyzed how the empirical generalization performance of GNNs behaves when we change several parameters such as miss alignment value of labels, graph size, and the number of observed points. This paper confirmed that the empirical generalization performance is consistent with theoretically derived generalization bounds.
  - [So1.2] In addition, the generalization bounds explain the benefit of the normalization of adjacency matrices.
- [So2] I have a question about the soundness of the experiment in Figure 1. The motivation for the TRC-based generalization bounds was that the VC-dimension-based bounds could be vacuous (l.179). However, in Figure 1, the theoretical bounds are outside (0, 1) (l.277). It raised the question of whether TRC bounds can truly also overcome the shortcomings of the VC-dimension-based bounds.

### Correctness (Are derivation of theorems and experiments correct?)

- [C1] I have questions about the correctness of Theorem 3 and Corollary 1 due to the lack of proof. In particular, I could not understand how I can derive Corollary 1. Since Theorem 3 derived only the "upper bound" of the generalization bound, it is unclear how we can compare generalization errors.

### Novelty and Significance (Do the paper have novel points? If so, are they significant?)

- [N1] The derivation is based on the theory of TRC in [EY09]. The central part of the proofs was to evaluate TRC. I would say that the proof did not have much technical novelty because it is a combination of characteristics of TRC. We can prove the characteristics similar to those of the usual Rademacher complexity.
- [N2] In addition, since [Oono and Suzuki, 20] derived the TRC-based generalization bounds for GNNs
- [N3] However, this paper differs from [Oono and Suzuki, 20] in the following three points. First, [Oono and Suzuki, 20] analyzed multi-scale GNNs, while the target of this paper was single-scale GNNs with possibly residual connections. Second, [Oono and Suzuki, 20] evaluated the SX norm but did not go further than that. On the contrary, this paper explained the effect of normalization by evaluating its order (eq. 10, 11). Third, this paper derived the dependence of generalization bound for the Planted model on the cluster-label miss-alignment ratio (Theorem 2). In addition, this paper verified the dependence experimentally. Considering these points, although these derivations are a combination of classical tools, I can recognize the significance of the results of this paper.

[Oono and Suzuki, 20] https://proceedings.neurips.cc/paper/2020/hash/dab49080d80c724aad5ebf158d63df41-Abstract.html

### Clarity (Is the paper clearly written?)

- [Cl1] The organization of this paper is OK. I can understand this paper without much difficulty.


### Other Comments

- l.54: derive→derives
- l.55: provide→provides
- l.79: This paper wrote that the situation considered in Section 2.2 is a "realistic case." However, I think it is somewhat difficult to say that the Erdos-Renyi graph and SBM are realistic. That being said, I agree with this paper in that being vacuous is a problem of the VC dimension-based bounds.
- l.105 I have a question about the problem setting. When this paper described the transductive learning setting, it took over the notation from the supervised learning setting. Therefore, it looks to me at first sight that we i.i.d. draw $m$ points from $\mathcal{D}_X = {\rm Unif}[n]$. However, it is not true as we draw data points without replacement in a transductive setting. So, I would like to suggest to reconsider the description.
- l.249: μ should be capitalized.
- Figure 1: In the figure, 90% quantile is shown. Since the experiment was run 5 times, does it mean that 90 % quantile is in fact the second value from the top?
- Figure 1 (Bottom center): In Cora, the sample size n is varied.
- l.256: In the following we can show ... → In the following, we can show ...
- l.258: loos → lose
- l.261: I think the notation $\rho=O(p), O(q)$ is not appropriate. Does it mean $\rho=O(\min(p, q))$ ?
- l.285: vary → verify
- l.299: Cite the over-smoothing effect, as it was coined by existing works e.g., [Li et al., AAAI2019].

[Li et al., AAAI2019] https://aaai.org/ocs/index.php/AAAI/AAAI18/paper/view/16098


### Post-rebuttal comments

I thank the authors for answering my questions sincerely. The approach of this paper is classical as the main tool is the (transductive version of) Rademacher complexity. Therefore, the novelty is limited in that respect (cf. [N1] of my review comment.) Considering that, I keep my score. However, However, since I acknowledge the significance of the conclusions drawn from the analysis (cf. [N3] of my review comment), I will not oppose to accepting the paper.


**Time Spent Reviewing:**

8

---

> ### Author Response · Authors · 2021-08-09
> **Response to reviewer’s comments**
>
> We thank the reviewer for the recommendations for the paper and also for the constructive feedback.
>
> In the following, we address open questions posed in the review.
>
> [So2] Generalisation error bounds, even for simple ML models, can exceed $1$ due to absolute constants that cannot be precisely estimated. Hence, the point of interest is the dependence of key parameters; for instance, in a supervised setting, the bounds are $O(1/\sqrt{m})$ and typically exceeds $1$ for moderate $m$. The main shortcomings of the derived bound for VC Dimension (see Proposition 1) is that the dependencies on the graph and feature information are non-informative. We noted, that even under strong assumptions on the graph, for example under consideration of Erdos-Renyi graphs or stochastic block models $\operatorname{rank}(\boldsymbol{S})=O(n)$ [CV08] holds. Therefore changes in the graph structure or feature information are not reflected in the VC Dimension bound. This is the main notion of ‘limitation’ that we consider. For the TRC bound, we see in Theorem 1 and Theorem 2 that this is no longer the case as we can explicitly analyze, e.g., the influence of graph information and the number of observed points. We, therefore, argue that TRC based bound can overcome the shortcomings of the VC Dimension based bound. As noted in the paper, the absolute value of the bound can be out of the $(0,1)$ range. While this does not allow us to numerically show how tight the bound is in practice, we can still make statements about the influence of the change of parameters, where the experiments validate the constancy between theory and empirical observations.
>
> [C1] Regarding the proof of Theorem 3, we would like to refer to section’ D.1 TRC bound on the Residual connection’ in the supplementary material, which is generally along the lines of the proof for Theorem 1 and some additional applications of basic TRC properties. With regards to Corollary 1, the reviewer’s observation is correct that the comparison is with respect to the upper bounds, and not the exact generalization gaps. We will remove eq. 14 and rephrase the corollary as a discussion in the revised version where we argue that the generalization error bound for GNNs is higher than the one for residual GNNs (and similarly for different $\alpha$).
>
> Other Comments:
>
> * l.79: We agree with the reviewer and will rephrase this point for the revised version to make it more clear. Our motivation was to argue that VC bounds are too pessimistic even under simple graph models.
>
> * Figure 1: Yes, it can be understood as stated by the reviewer. While we only plot the results for five runs in the paper, partial experiments on a larger number of initialization showed the same behavior.
>
> * All other textual/minor comments we will address in the revised version.
>
> [CV08] Kevin P. Costello and Van H. Vu. “The rank of random graphs”. In: Random Structures \& Algorithms. 2008.
>
> [EYP09] Ran El-Yaniv and Dmitry Pechyony. “Transductive Rademacher Complexity and its Applications”. In: Journal of Artificial Intelligence Research. 2009.

---

> > ### Comment · Reviewer_d7vj · 2021-08-26
> > **Thank you for your responses.**
> >
> > I thank the authors for considering my review comments sincerely and answering my questions.
> >
> > - [So2] I understand that the TRC gives better analyses that reflect graph and feature structures in a finer resolution compared with the VC dimension. On the other hand, I also understand that the TRC-based analysis did not solve the problem of vacuous upper bounds similarly to the VC-based analysis.
> > - [C1] I am sorry that I did not notice that Section D has proof for Theorem 3. I agree with the authors to rephrase the collorary as a discussion.
> > - l.79: OK
> > - Figure 1: OK

---

> > > ### Author Response · Authors · 2021-09-02
> > > **Clarification on vacuous upper bounds from TRC**
> > >
> > > We thank the reviewer for the response. The comment in the response about “vacuous upper bounds” probably arises from a misinterpretation of the discussion in the paper. We assume that, by being vacuous, the reviewer refers to the fact that the bound exceeds 1. Our comment in l.179 that VC-bound is trivial is not simply because it exceeds 1, but the fact that the VC bound cannot capture the structure of the graph (we will rephrase this statement).
> > >
> > > We note that existing works on generalisation bounds for GCN do not empirically validate that their bounds follow similar trends as test error, which we could using TRC bounds.
> > >
> > > It is also worth noting that for simple models (decision stumps or linear classifiers), where VC bounds are meaningful, the bounds still exceed 1 for small sample sizes due to the effect of absolute constants in the bound. Even in such simple models, the sample size need to be quite large so that the bound is below 1. Yet VC-bounds are useful there since they explain trends in test error. The message of this paper is that, for GCNs, VC-bounds do not explain trends but TRC bounds do. To highlight this, we will include plots for VC bounds (rescaled).

---

### Official Review · Reviewer_16pJ · 2021-07-20

**Rating:** 7
**Confidence:** 2

**Summary:**

The paper studies generalisation bounds of GNNs from the classical statistical learning point of view. Under some specific distribution assumptions, the authors show that, while VC Dimension does result in trivial generalisation error bounds,  transductive Rademacher complexity can explain the generalisation properties of graph convolutional networks for the specific case of stochastic block models. Experimental results on synthetic data generated from SBM graphs, and the CORA dataset quantify the validity of the bound. Finally, the influence of depth and residual connections is studied.

**Ethical Concerns:**

I do not see important ethical concerns with this paper.

**Limitations And Societal Impact:**

The authors clearly state the limitations of their framework in the conclusion part. One additional limitation is the extension to weighted graphs and more labels (not simply binary classification).

**Main Review:**

This is a very interesting and well-presented paper, with respect to different aspects.

Originality: Generalisation bounds of GNNs are definitely worth investigating, and it is a topic of great interest to the research community. The proposed approach is relevant and interesting. Specifically, the authors focus on transduction settings that have been largely underlooked in the literature. Studying Rademacher complexity seems to provide some insight on the generalisation properties, at least on stochastic block models.

Quality: The paper is of very good quality, and technically sound. Proofs are provided. Experimental results are informative. A more extensive evaluation on different type of graph models and well as different feature distribution would significantly increase the contribution.

Clarity: The paper is well written. Despite being a heavy and very technical paper, the authors did a good job in terms of the presentation of the results. A few comments are the following:

Typos:
Line 50: is on 'a' learning: remove 'a'
Line 58: deep learning 'learning' repeated twice
Line 112: to be instead of is
Line 145: 'can' is repeated twice
Line 198: it suggests instead of suggest
Figure 1: first subfigure: alignment instead of allignment
Line 256: loose instead of loos
Line 294: remove a
Line 311: more than

Explain a bit definition 1.
Remind what K is in the conclusions.

Significance: The obtain results are theoretically interesting and I expect them to generate more research in the topic. Although the current results are specific to some cases of graph models and feature distributions, they are definitely relevant and a useful step towards understanding theoretically GNNs. I would expect the authors to extend the bounds to terms that are related to the graph characteristics/distribution, as well as some more interpretable features.

After rebuttal:
The authors should include a more extensive discussion about the existing literature (e.g.,Oono&Suzuki (2020)) and position their work with respect to that.


**Time Spent Reviewing:**

5

---

> ### Author Response · Authors · 2021-08-09
> **Response to reviewer’s comments**
>
> We thank the reviewer for the positive evaluation of the paper and also for the constructive feedback with regards to follow-up work and extensions. The analysis of further different types of graph models, as well as different feature distributions, will be added as a future work direction in the revised version.
>
> Textual/minor comments we will address in the revised version.

---

> > ### Comment · Reviewer_16pJ · 2021-09-02
> > **After rebuttal**
> >
> > I thank the authors for their response. While I still believe that this is a good paper, I share the concerns of the other reviewers about a more clear positioning of the work with respect to the state of the art. I would expect the authors to clarify that part in the final version.

---

### Decision · Program_Chairs · 2021-09-27

**Decision:**

Accept (Poster)

**Comment:**

This paper analyzed the generalization gap of GNNs based on statistical learning theory. First, this paper gives an upper bound of the generalization gap by using the VC dimension of the model, which is characterized by the rank of the adjacency matrix. However, it is shown that such a bound is vacuous. To overcome this issue, the authors give a tighter Transductive Rademacher Complexity (TRC) which is bounded by norms of $S$ and $SX$. They gave detailed discussions about the derived bound in some specific situations. They also discussed benefit of residual connections which reduce the generalization gap.

Overall, this paper is well written, and the problems that the paper is analyzing is indeed important. This paper gives a detailed analysis on the issue of generalization ability of graph neural networks, which is valuable to the community.
One of the biggest weakness of this study is that evaluating TRC is not new because it has already been addressed by Oono and Suzuki (2020). Unfortunately, this paper does not cite Oono and Suzuki (2020) nor give discuss relation to it. On the other hand, there are several interesting insights that were not indicated by Oono and Suzuki (2020). In that sense, this paper has novelty.

In summary, this paper gives an instructive theoretical analysis for GNNs and it can be accepted by NeurIPS.
On the other hand, I definitely recommend the authors to include Oono and Suzuki (2020) and discuss relation to it.